# A Statistical Analysis of Wasserstein Autoencoders for Intrinsically Low-dimensional Data

**Saptarshi Chakraborty**
UC Berkeley
saptarshic@berkeley.edu

**Peter L. Bartlett**
Google DeepMind & UC Berkeley
peter@berkeley.edu

## Abstract

Variational Autoencoders (VAEs) have gained significant popularity among researchers as a powerful tool for understanding unknown distributions based on limited samples. This popularity stems partly from their impressive performance and partly from their ability to provide meaningful feature representations in the latent space. Wasserstein Autoencoders (WAEs), a variant of VAEs, aim to not only improve model efficiency but also interpretability. However, there has been limited focus on analyzing their statistical guarantees. The matter is further complicated by the fact that the data distributions to which WAEs are applied - such as natural images - are often presumed to possess an underlying low-dimensional structure within a high-dimensional feature space, which current theory does not adequately account for, rendering known bounds inefficient. To bridge the gap between the theory and practice of WAEs, in this paper, we show that WAEs can learn the data distributions when the network architectures are properly chosen. We show that the convergence rates of the expected excess risk in the number of samples for WAEs are independent of the high feature dimension, instead relying only on the intrinsic dimension of the data distribution.

## 1 Introduction

The problem of understanding and possibly simulating samples from an unknown distribution only through some independent realization of the same is a key question for the machine learning community. Parallelly with the appearance of Generative Adversarial Networks (GANs) (Goodfellow et al., 2014), Variational Autoencoders (Kingma & Welling, 2014) have also gained much attention not only due to their useful feature representation properties in the latent space but also for data generation capabilities. It is important to note that in GANs, the generator network learns to create new samples that are similar to the training data by fooling the discriminator network. However, GANs and their popular variants do not directly provide a way to manipulate the generated data or explore the latent space of the generator. On the other hand, a VAE learns a latent space representation of the input data and allows for interpolation between the representations of different samples. Several variants of VAEs have been proposed to improve their generative performance. One popular variant is the conditional VAE (CVAE) (Sohn et al., 2015), which adds a conditioning variable to the generative model and has shown remarkable empirical success. Other variants include InfoVAE (Zhao et al., 2017), $\beta$-VAE (Higgins et al., 2017), and VQ-VAE (Van Den Oord et al., 2017), etc., which address issues such as disentanglement, interpretability, scalability, etc. Recent works have shown the effectiveness of VAEs and their variants in a variety of applications, including image (Gregor et al., 2015) and text generation (Yang et al., 2017), speech synthesis (Tachibana et al., 2018), and drug discovery (Gómez-Bombarelli et al., 2018). A notable example is the DALL-E model (Ramesh et al., 2021), which uses a VAE to generate images from textual descriptions.

However, despite their effectiveness in unsupervised representation learning, VAEs have been heavily criticized for their poor performance in approximating multi-modal distributions. Influenced by the superior performance of GANs, researchers have attempted to leverage this advantage of adversarial losses by incorporating them into VAE objective (Makhzani et al., 2016; Mescheder et al., 2017). Wasserstein Autoencoder (WAEs) (Tolstikhin et al., 2018) tackles the problem from an opti-

mal transport viewpoint. Incorporating such a GAN-like architecture, not only preserves the latent space representation that is unavailable in GANs but also enhances data generation capabilities. Both VAEs and WAEs attempt to minimize the sum of a reconstruction cost and a regularizer that penalizes the difference between the distribution induced by the encoder and the prior distribution on the latent space. While VAEs force the encoder to match the prior distribution for each input example, which can lead to overlapping latent codes and reconstruction issues, WAEs force the continuous mixture of the encoder distribution over all input examples to match the prior distribution, allowing different examples to have more distant latent codes and better reconstruction results. Furthermore, the use of the Wasserstein distance allows WAEs to incorporate domain-specific constraints into the learning process. For example, if the data is known to have a certain structure or topology, this information can be used to guide the learning process and improve the quality of generated samples. This results in a more robust model that can handle a wider range of distributions, including multimodal and heavy-tailed distributions.

While VAE and its variants have demonstrated empirical success, little attention has been given to analyzing their statistical properties. Recent developments from an optimization viewpoint include Rolinek et al. (2019), who showed VAEs pursue Principal Component Analysis (PCA) embedding under certain situations, and Koehler et al. (2022), who analyzed the implicit bias of VAEs under linear activation with two layers. For explaining generalization, Tang & Yang (2021) proposed a framework for analyzing excess risk for vanilla VAEs through M-estimation. When having access to $n$ i.i.d. samples from the target distribution, Chakrabarty & Das (2021) derived a bound based on the Vapnik-Chervonenkis (VC) dimension, providing a guarantee of $\mathcal{O}(n^{-1/2})$-convergence with a non-zero margin of error, even under model specification. However, their analysis is limited to a parametric regime under restricted assumptions and only considers a theoretical variant of WAEs, known as $f$-WAEs (Husain et al., 2019), which is typically not implemented in practice.

Despite recent advancements in the understanding of VAEs and their variants, existing analyses fail to account for the fundamental goal of these models, i.e. to understand the data generation mechanism where one can expect the data to have an intrinsically low-dimensional structure. For instance, a key application of WAEs is to understand natural image generation mechanisms and it is believed that natural images have a low-dimensional structure, despite their high-dimensional pixel-wise representation (Pope et al., 2020). Furthermore, the current state-of-the-art views the problem only through a classical learning theory approach to derive $\mathcal{O}(n^{-1/2})$ or faster rates (under additional assumptions) ignoring the model misspecification error. Thus, such rates do not align with the well-known rates for classical non-parametric density estimation approaches (Kim et al., 2019). Additionally, these approaches only consider the scenario where the network architecture is fixed, but in practice, larger models are often employed for big datasets.

In this paper, we aim to address the aforementioned shortcomings in the current literature and bridge the gap between the theory and practice of WAEs. Our contributions include:

- We propose a framework to provide an error analysis of Wasserstein Autoencoders (WAEs) when the data lies in a low-dimensional structure in the high-dimensional representative feature space.

- Informally, our results indicate that if one has $n$ independent and identically distributed (i.i.d.) samples from the target distribution, then under the assumption of Lipschitz-smoothness of the true model, if the corresponding networks are properly chosen, the error rate for the problem scales as $\tilde{\mathcal{O}}\left(n^{-\frac{1}{2+d_\mu}}\right)$, where, $d_\mu$ is the upper Minkowski dimension of the support of the target distribution.

- The networks can be chosen as having $\mathcal{O}(n^{\gamma_e})$ many weights for the encoder and $\mathcal{O}(n^{\gamma_g})$ for the generator, where, $\gamma_e, \gamma_g \leq 1$ and only depend on $d_\mu$ and $\ell$ (dimension of the latent space), respectively. Furthermore, the values of $\gamma_e$ and $\gamma_g$ decrease as the true model becomes smoother.

- We show that one can ensure encoding and decoding guarantees, i.e. the encoded distribution is close enough to the target latent distribution, and the generator maps back the encoded points close to the original points. Under additional regularity assumptions, we show that the approximating push-forward measure, induced by the generator, is close to the target distribution, in the Wasserstein sense, almost surely.

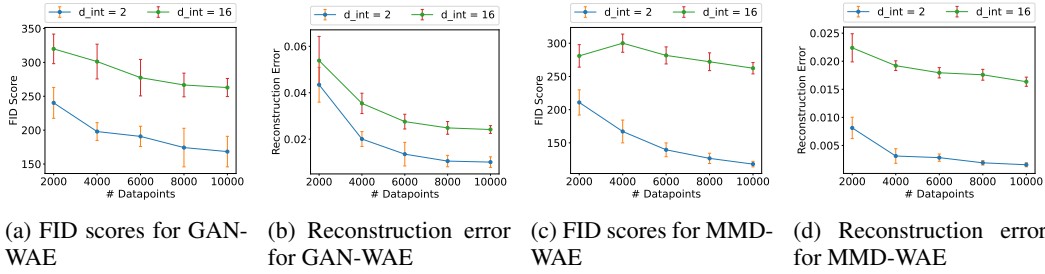

(a) FID scores for GAN-WAE

(b) Reconstruction error for GAN-WAE

(c) FID scores for MMD-WAE

(d) Reconstruction error for MMD-WAE

Figure 1: Average generalization error (in terms of FID scores) and reconstruction test errors for different values of $n$ for GAN and MMD variants of WAE. The error bars denote the standard deviation out of 10 replications.

## 2 A PROOF OF CONCEPT

Before we theoretically explore the problem, we discuss an experiment to demonstrate that the error rates for WAEs depend primarily only on the intrinsic dimension of the data. Since it is difficult to assess the intrinsic dimensionality of natural images, we follow the prescription of Pope et al. (2020) to generate low-dimensional synthetic images. We use a pre-trained Bi-directional GAN (Donahue et al., 2017) with 128 latent entries and outputs of size $128 \times 128 \times 3$, trained on the ImageNet dataset (Deng et al., 2009). Using the decoder of this pre-trained BiGAN, we generate $11,000$ images, from the class, `soap-bubble` where we fix most entries of the latent vectors to zero leaving only `d_int` free entries. We take `d_int` to be 2 and 16, respectively. We reduce the image sizes to $28 \times 28$ for computational ease. We train a WAE model with the standard architecture as proposed by Tolstikhin et al. (2018) with the number of training samples varying in $\{2000, 4000, \ldots, 10000\}$ and keep the last 1000 images for testing. For the latent distribution, we use the standard Gaussian distribution on the latent space $\mathbb{R}^8$ and use the Adam optimizer (Kingma & Ba, 2015) with a learning rate of 0.0001. We also take $\lambda = 10$ for the penalty on the dissimilarity in objective (4). After training for 10 epochs, we generate 1000 sample images from the distribution $\hat{G}_\sharp \nu$ (see Section 3 for notations) and compute the Frechet Inception Distance (FID) (Heusel et al., 2017) to assess the quality of the generated samples with respect to the target distribution. We also compute the reconstruction error for these test images. We repeat the experiment 10 times and report the average. The experimental results for both variants of WAE, i.e. the GAN and MMD are shown in Fig. 1. It is clear from Fig. 1 that the error rates for `d_int` = 2 is lower than for the case `d_int` = 16. The codes for this experimental study can be found at `https://github.com/SaptarshiC98/WAE`.

## 3 BACKGROUND

### 3.1 NOTATIONS AND SOME PRELIMINARY CONCEPTS

This section introduces preliminary notation and concepts for theoretical analyses.

**Notation**  We use notations $x \vee y := \max\{x, y\}$ and $x \wedge y := \min\{x, y\}$. $T_\sharp \mu$ denotes the push-forward of measure $\mu$ by the map $T$. For function $f : \mathcal{S} \to \mathbb{R}$, and probability measure $\gamma$ on $\mathcal{S}$, let $\|f\|_{\mathbb{L}_p(\gamma)} := \left( \int_{\mathcal{S}} |f(x)|^p d\gamma(x) \right)^{1/p}$. Similarly, $\|f\|_{\mathbb{L}_\infty(\mathcal{A})} := \sup_{x \in \mathcal{A}} |f(x)|$. For any function class $\mathcal{F}$, and distributions $P$ and $Q$, $\|P - Q\|_{\mathcal{F}} = \sup_{f \in \mathcal{F}} |\int f dP - \int f dQ|$ denotes the Integral Probability Metric (IPM) w.r.t. $\mathcal{F}$. We say $A_n \lesssim B_n$ (also written as $A_n = \mathcal{O}(B_n)$) if there exists $C > 0$, independent of $n$, such that $A_n \leq C B_n$. Similarly, we use the notation, $A_n \stackrel{\sim}{\lesssim} B_n$ (also written as $A_n = \tilde{\mathcal{O}}(B_n)$) if $A_n \leq C B_n \log^C(en)$, for some $C > 0$. We say $A_n \asymp B_n$, if $A_n \lesssim B_n$ and $B_n \lesssim A_n$. For a function $f : \mathbb{R}^{d_1} \to \mathbb{R}^{d_2}$, we write, $\|f\|_{\text{Lip}} = \sup_{x \neq y} \frac{\|f(x) - f(y)\|_2}{\|x - y\|_2}$.

**Definition 1** (Neural networks)**.**  Let $L \in \mathbb{N}$ and $\{N_i\}_{i \in [L]} \subset \mathbb{N}$. Then a $L$-layer neural network $f : \mathbb{R}^d \to \mathbb{R}^{N_L}$ is defined as,

$$f(x) = A_L \circ \sigma_{L-1} \circ A_{L-1} \circ \cdots \circ \sigma_1 \circ A_1(x) \tag{1}$$

Here, $A_i(y) = W_i y + b_i$, with $W_i \in \mathbb{R}^{N_i \times N_{i-1}}$ and $b_i \in \mathbb{R}^{N_{i-1}}$, with $N_0 = d$. $\sigma_j$ is applied component-wise. Here, $\{W_i\}_{1 \leq i \leq L}$ are known as weights, and $\{b_i\}_{1 \leq i \leq L}$ are known as biases. $\{\sigma_i\}_{1 \leq i \leq L-1}$ are known as the activation functions. Without loss of generality, one can take $\sigma_\ell(0) =$

$0, \forall \ell \in [L-1]$. We define the following quantities: (Depth) $\mathcal{L}(f) := L$ is known as the depth of the network; (Number of weights) The number of weights of the network $f$ is denoted as $\mathcal{W}(f)$.

$$\mathcal{NN}_{\{\sigma_i\}_{i \in [L-1]}}(L, W, R) = \{f \text{ of the form (1)} : \mathcal{L}(f) \leq L, \mathcal{W}(f) \leq W, \sup_{\boldsymbol{x} \in \mathbb{R}^d} \|f(\boldsymbol{x})\|_{\infty} \leq R\}.$$

If $\sigma_j(x) = x \vee 0$, for all $j = 1, \ldots, L-1$, we denote $\mathcal{NN}_{\{\sigma_i\}_{1 \leq i \leq L-1}}(L, W, R)$ as $\mathcal{RN}(L, W, R)$. We often omit $R$ in cases where it is clear that $R$ is bounded by a constant.

**Definition 2** (Hölder functions). Let $f : \mathcal{S} \to \mathbb{R}$ be a function, where $\mathcal{S} \subseteq \mathbb{R}^d$. For a multi-index $\boldsymbol{s} = (s_1, \ldots, s_d)$, let, $\partial^{\boldsymbol{s}} f = \frac{\partial^{|\boldsymbol{s}|} f}{\partial x_1^{s_1} \ldots \partial x_d^{s_d}}$, where, $|\boldsymbol{s}| = \sum_{\ell=1}^d s_\ell$. We say that a function $f : \mathcal{S} \to \mathbb{R}$ is $\beta$-Hölder (for $\beta > 0$) if

$$\|f\|_{\mathcal{H}^{\beta}} := \sum_{\boldsymbol{s}:0 \leq |\boldsymbol{s}| < \lfloor \beta \rfloor} \|\partial^{\boldsymbol{s}} f\|_{\infty} + \sum_{\boldsymbol{s}:|\boldsymbol{s}|=\lfloor \beta \rfloor} \sup_{x \neq y} \frac{\|\partial^{\boldsymbol{s}} f(x) - \partial^{\boldsymbol{s}} f(y)\|}{\|x-y\|^{\beta - \lfloor \beta \rfloor}} < \infty.$$

If $f : \mathbb{R}^{d_1} \to \mathbb{R}^{d_2}$, then we define $\|f\|_{\mathcal{H}^{\beta}} = \sum_{j=1}^{d_2} \|f_j\|_{\mathcal{H}^{\beta}}$.

For notational simplicity, let, $\mathcal{H}^{\beta}(\mathcal{S}_1, \mathcal{S}_2, C) = \{f : \mathcal{S}_1 \to \mathcal{S}_2 : \|f\|_{\mathcal{H}^{\beta}} \leq C\}$. Here, both $\mathcal{S}_1$ and $\mathcal{S}_2$ are both subsets of a real vector spaces.

**Definition 3** (Maximum Mean Discrepancy (MMD)). Let $\mathbb{H}_{\mathcal{K}}$ be the Reproducible Kernel Hilbert Space (RKHS) corresponding to the reproducing kernel $\mathcal{K}(\cdot, \cdot)$, defined on $\mathbb{R}^d$. Let the corresponding norm in this RKHS be $\| \cdot \|_{\mathbb{H}_{\mathcal{K}}}$. The Maximum Mean Discrepancy between two distributions $P$ and $Q$ is defined as: $\text{MMD}_{\mathcal{K}}(P, Q) = \sup_{f:\|f\|_{\mathbb{H}_{\mathcal{K}}} \leq 1} \left( \int f dP - \int f dQ \right)$.

### 3.2 WASSERSTEIN AUTOENCODERS

Let $\mu$ be a distribution in the data-space $\mathcal{X} = [0,1]^d$ and $\mathcal{Z} = [0,1]^{\ell}$ be the latent space. In Wasserstein Autoencoders (Tolstikhin et al., 2018), one tries to learn a generator map, $G : \mathcal{Z} \to \mathcal{X}$ and an encoder map $E : \mathcal{X} \to \mathcal{Z}$ by minimizing the following objective,

$$V(\mu, \nu, G, E) = \int c(x, G \circ E(x)) d\mu(x) + \lambda \text{diss}(E_{\sharp}\mu, \nu). \tag{2}$$

Here, $\lambda > 0$ is a hyper-parameter, often tuned based on the data. The first term in (2) aims to minimize a reconstruction error, i.e. the decoded value of the encoding should approximately result in the same value. The second term ensures that the encoded distribution is close to a known distribution $\nu$ that is easy to sample from. The function $c(\cdot, \cdot)$-is a loss function on the data space. For example, Tolstikhin et al. (2018) took $c(x, y) = \|x - y\|_2^2$. $\text{diss}(\cdot, \cdot)$ is a dissimilarity measure between probability distributions defined on the latent space. Tolstikhin et al. (2018) recommended either a GAN-based dissimilarity measure or a Maximum Mean Discrepancy (MMD)-based measure (Gretton et al., 2012). In this paper, we will consider the special cases, where this dissimilarity measure is taken to be the Wasserstein-1 metric, which is the dissimilarity measure for WGANs (Arjovsky et al., 2017; Gulrajani et al., 2017) or the squared MMD-metric.

In practice, however, one does not have access to $\mu$ but only a sample $\{X_i\}_{i \in [n]}$, assumed to be independently generated from $\mu$. Let $\hat{\mu}_n$ be the empirical measure based on the data. One then minimizes the following empirical objective to estimate $E$ and $G$.

$$V(\hat{\mu}_n, \nu, G, E) = \int c(x, G \circ E(x)) d\hat{\mu}_n(x) + \lambda \widehat{\text{diss}}(E_{\sharp}\hat{\mu}_n, \nu). \tag{3}$$

Here, $\widehat{\text{diss}}(\cdot, \cdot)$ is an estimate of $\text{diss}(\cdot, \cdot)$, based only on the data, $\{X_i\}_{i \in [n]}$. For example, if $\text{diss}(\cdot, \cdot)$ is taken to be the Wasserstein-1 metric, then, $\widehat{\text{diss}}(E_{\sharp}\hat{\mu}_n, \nu) = \mathcal{W}_1(E_{\sharp}\hat{\mu}_n, \nu)$. On the other hand, if $\text{diss}(\cdot, \cdot)$ is taken to be the $\text{MMD}_{\mathcal{K}}^2$-measure, one can take,

$$\widehat{\text{diss}}(E_{\sharp}\hat{\mu}_n, \nu) = \frac{1}{n(n-1)} \sum_{i \neq j} \mathcal{K}(E(X_i), E(X_j)) + \mathbb{E}_{Z,Z' \sim \nu} \mathcal{K}(Z, Z') - \frac{2}{n} \sum_{i=1}^n \int \mathcal{K}(E(X_i), z) d\nu(z).$$

Of course, in practice, one does a further estimation of the involved dissimilarity measure through taking an estimate $\hat{\nu}_m$, based on $m$ i.i.d samples $\{Z_j\}_{j \in [m]}$ from $\nu$, i.e. $\hat{\nu}_m = \frac{1}{m} \sum_{j=1}^m \delta_{Z_j}$. In this case the estimate of $V$ in (2) is given by,

$$V(\hat{\mu}_n, \hat{\nu}_m, G, E) = \int c(x, G \circ E(x)) d\hat{\mu}_n(x) + \lambda \widehat{\text{diss}}(E_{\sharp}\hat{\mu}_n, \nu_m). \tag{4}$$

If $\text{diss}(\cdot, \cdot)$ is taken to be the Wasserstein-1 metric, then, $\widehat{\text{diss}}(E_\sharp \hat{\mu}_n, \hat{\nu}_m) = \mathcal{W}_1(E_\sharp \hat{\mu}_n, \hat{\nu}_m)$. On the other hand, if $\text{diss}(\cdot, \cdot)$ is taken to be the $\text{MMD}_{\mathcal{K}}^2$-measure, one can take,

$$\widehat{\text{diss}}(E_\sharp \hat{\mu}_n, \hat{\nu}_m) = \frac{1}{n(n-1)} \sum_{i \neq j} \mathcal{K}(E(X_i), E(X_j)) + \frac{1}{m(m-1)} \sum_{i \neq j} \mathcal{K}(Z_i, Z_j) - \frac{2}{nm} \sum_{i=1}^{n} \sum_{j=1}^{m} \mathcal{K}(E(X_j), Z_j).$$

Suppose that $\Delta_{\text{opt}} > 0$ is the optimization error. The empirical WAE estimates satisfy the following properties:

$$(\hat{G}^n, \hat{E}^n) \in \left\{ G \in \mathcal{G}, E \in \mathcal{E} : V(\hat{\mu}_n, \nu, G, E) \leq \inf_{G \in \mathcal{G}, E \in \mathcal{E}} V(\hat{\mu}_n, \nu, G, E) + \Delta_{\text{opt}} \right\} \tag{5}$$

$$(\hat{G}^{n,m}, \hat{E}^{n,m}) \in \left\{ G \in \mathcal{G}, E \in \mathcal{E} : V(\hat{\mu}_n, \hat{\nu}_n, G, E) \leq \inf_{G \in \mathcal{G}, E \in \mathcal{E}} V(\hat{\mu}_n, \hat{\nu}_m, G, E) + \Delta_{\text{opt}} \right\}. \tag{6}$$

The functions in $\mathcal{G}$ and $\mathcal{E}$ are implemented through neural networks with ReLU activation $\mathcal{RN}(L_g, W_g)$ and $\mathcal{RN}(L_e, W_e)$, respectively.

## 4 INTRINSIC DIMENSION OF DATA DISTRIBUTION

Real data is often assumed to have a lower-dimensional structure within the high-dimensional feature space. Various approaches have been proposed to characterize this low dimensionality, with many using some form of covering number to measure the effective dimension of the underlying measure. Recall that the $\epsilon$-covering number of $\mathcal{S}$ w.r.t. the metic $\varrho$ is defined as $\mathcal{N}(\epsilon; \mathcal{S}, \varrho) = \inf\{n \in \mathbb{N} : \exists x_1, \ldots x_n \text{ such that } \cup_{i=1}^{n} B_\varrho(x_i, \epsilon) \supseteq \mathcal{S}\}$, with $B_\varrho(x, \epsilon) = \{y : \varrho(x, y) < \epsilon\}$. We characterize this low-dimensional nature of the data, through the (upper) Minkowski dimension of the support of $\mu$. We recall the definition of Minkowski dimensions,

**Definition 4** (Upper Minkowski dimension). *For a bounded metric space $(\mathcal{S}, \varrho)$, the upper Minkwoski dimension of $\mathcal{S}$ is defined as $\overline{\dim}_M(\mathcal{S}, \varrho) = \limsup_{\epsilon \to 0} \frac{\log \mathcal{N}(\epsilon; \mathcal{S}, \varrho)}{\log(1/\epsilon)}$.*

Throughout this analysis, we will assume that $\varrho$ is the $\ell_\infty$-norm and simplify the notation to $\overline{\dim}_M(\mathcal{S})$. $\overline{\dim}_M(\mathcal{S}, \varrho)$ essentially measures how the covering number of $\mathcal{S}$ is affected by the radius of balls covering that set. As the concept of dimensionality relies solely on covering numbers and doesn't require a smooth mapping to a lower-dimensional Euclidean space, it encompasses both smooth manifolds and even highly irregular sets like fractals. In the literature, Kolmogorov & Tikhomirov (1961) provided a comprehensive study on the dependence of the covering number of different function classes on the underlying Minkowski dimension of the support. Nakada & Imaizumi (2020) showed how deep regression learners can incorporate this low-dimensionality of the data that is also reflected in their convergence rates. Recently, Huang et al. (2022) showed that WGANs can also adapt to this low-dimensionality of the data. For any measure $\mu$ on $[0, 1]^d$, we use the notation $d_\mu := \overline{\dim}_M(\text{supp}(\mu))$. When the data distribution is supported on a low-dimensional structure in the nominal high-dimensional feature space, one can expect $d_\mu \ll d$.

It can be observed that the image of a unit hypercube under a Hölder map has a Minkowski dimension that is no more than the dimension of the pre-image divided by the exponent of the Hölder map.

**Lemma 5.** *Let, $f \in \mathcal{H}^\gamma\left(\mathcal{A}, [0, 1]^{d_2}, C\right)$, with $\mathcal{A} \subseteq [0, 1]^{d_1}$. Then, $\overline{\dim}_M\left(f\left(\mathcal{A}\right)\right) \leq \overline{\dim}_M(\mathcal{A})/(\gamma \wedge 1)$.*

## 5 THEORETICAL ANALYSES

### 5.1 ASSUMPTIONS AND ERROR DECOMPOSITION

To facilitate theoretical analysis, we assume that the data distributions are realizable, meaning that a "true" generator and a "true" encoder exist. Specifically, we make the assumption that there is a true smooth encoder that maps the $\mu$ to $\nu$, and the left inverse of this true encoder exists and is also smooth. Formally,

**A1.** *There exists $\tilde{G} \in \mathcal{H}^{\alpha_g}([0, 1]^d, [0, 1]^\ell, C)$ and $\tilde{E} \in \mathcal{H}^{\alpha_e}([0, 1]^\ell, [0, 1]^d, C)$, such that, $\tilde{E}_\sharp \mu = \nu$ and $(\tilde{G} \circ \tilde{E})(\cdot) = id(\cdot)$, a.e. $[\mu]$.*

It is also important to note that A1 entails that the manifold has a single chart, in a probabilistic sense, which is a strong assumption. Naturally, when it comes to GANs, one can work with a weaker assumption as the learning task becomes notably much simpler as one does not have to learn an inverse map to the latent space. A similar problem, while analyzing autoencoders, was faced by Liu et al. (2023) where they tackled the problem by considering chart-autoencoders, which have additional components in the network architecture, compared to regular autoencoders. A similar approach of employing chart-based WAEs could be proposed and subjected to rigorous analysis. This potential avenue could be an intriguing direction for future research.

One immediate consequence of assumption A1 ensures that the generator maps $\nu$ to the target $\mu$. We can also ensure that the latent distribution remains unchanged if one passes it through the generator and maps it back through the encoder. Furthermore, the objective function (2) at this true encoder-generator pair takes the value, zero, as expected.

**Proposition 6.** *Under assumption A1, the following holds:* $(a)\,\tilde{G}_{\sharp}\nu = \mu$, $(b)\,(\tilde{E}\circ\tilde{G})_{\sharp}\nu = \nu$, $(c)\,V(\mu,\nu,\tilde{G},\tilde{E}) = 0$.

From Lemma 5, It is clear that $d_{\mu} = \overline{\dim}_M(\text{supp}(\mu)) \leq \overline{\dim}_M\left(\tilde{G}\left([0,1]^{\ell}\right)\right) \leq \max\{\ell/(\alpha_g\wedge 1),d\}$. If $\ell \ll d$ and $\alpha_g$ is not very small, then, $d_{\mu} = (\alpha_g\wedge 1)^{-1}\ell \ll d$. Thus, the usual conjecture of $d_{\mu} \ll d$ can be modeled through assumption A1 when the latent space has a much smaller dimension and the true generator is well-behaved, i.e. $\alpha_g$ is not too small.

A key step in the theoretical analysis is the following oracle inequality that bounds the excess risk in terms of the optimization error, misspecification error, and generalization error.

**Lemma 7** (Oracle Inequality). *Suppose that,* $\mathcal{F} = \{f(x) = c(x,G\circ E(x)) : G \in \mathcal{G},\ E \in \mathcal{E}\}$. *Then the following hold:*

$$V(\mu,\nu,\hat{G}^n,\hat{E}^n) \leq \Delta_{miss} + \Delta_{opt} + 2\|\hat{\mu}_n - \mu\|_{\mathcal{F}} + 2\lambda \sup_{E\in\mathcal{E}}|\widehat{diss}(E_{\sharp}\hat{\mu}_n,\nu) - diss(E_{\sharp}\mu,\nu)|. \tag{7}$$

$$V(\mu,\nu,\hat{G}^{n,m},\hat{E}^{n,m}) \leq \Delta_{miss} + \Delta_{opt} + 2\|\hat{\mu}_n - \mu\|_{\mathcal{F}} + 2\lambda \sup_{E\in\mathcal{E}}|\widehat{diss}(E_{\sharp}\hat{\mu}_n,\hat{\nu}_m) - diss(E_{\sharp}\mu,\nu)|. \tag{8}$$

*Here,* $\Delta_{miss} = \inf_{G\in\mathcal{G},E\in\mathcal{E}} V(\mu,\nu,G,E)$ *denotes the misspecification error for the problem.*

For our theoretical analysis, we need to ensure that the used kernel in the MMD and the loss function $c(\cdot,\cdot)$ are regular enough. To impose such regularity, we assume the following:

**A2.** *We assume that, (a) for some* $B > 0$, $\mathcal{K}(x,y) \leq B^2$, *for all* $x,y \in [0,1]^{\ell}$; *(b) For some* $\tau_k$, $|\mathcal{K}(x,y) - \mathcal{K}(x',y')| \leq \tau_k(\|x-x'\|_2 + \|y-y'\|_2)$.

**A3.** *The loss function* $c(\cdot,\cdot)$ *is Lipschitz on* $[0,1]^d \times [0,1]^d$, *i.e.* $|c(x,y) - c(x',y')| \leq \tau_c(\|x-x'\|_2 + \|y-y'\|_2)$ *and* $c(x,y) \leq B_c$, *for all,* $x,y \in [0,1]^d$.

## 5.2 MAIN RESULT

Under assumptions A1–3, one can control the expected excess risk of the WAE problem for both the $\mathcal{W}_1$ and MMD dissimilarities. The main idea is to select appropriate sizes for the encoder and generator networks, that minimize both the misspecification errors and generalization errors to bound the expected excess risk using Lemma 7. Theorem 8 shows that one can appropriately select the network size in terms of the number of samples available, i.e $n$, to achieve a trade-off between the generalization and misspecification errors as selecting a larger network facilitates better approximation but makes the generalization gap wider and vice-versa. The main result of this paper is stated as follows.

**Theorem 8.** *Suppose that assumptions A1–3 hold and* $\Delta_{opt} \leq \Delta$ *for some fixed non-negative threshold* $\Delta$. *Furthermore, suppose that* $s > d_{\mu}$. *Then we can find* $n_0 \in \mathbb{N}$ *and* $\beta > 0$, *that might depend on* $d,\ell,\alpha_g,\alpha_e,\tilde{G}$ *and* $\tilde{E}$, *such that if* $n \geq n_0$, *we can choose* $\mathcal{G} = \mathcal{RN}(L_g,W_g)$ *and* $\mathcal{E} = \mathcal{RN}(L_e,W_e)$, *with,* $L_e \leq \beta\log n$, $W_e \leq \beta n^{\frac{s}{2\alpha_e+s}}\log n$, $L_g \leq \beta\log n$ *and* $W_g \leq \beta n^{\frac{\ell}{\alpha_e(\alpha_g\wedge 1)+\ell}}\log n$, *then, for the estimation problem* (5),

*(a)* $\mathbb{E}V(\mu,\nu,\hat{G}^n,\hat{E}^n) \lesssim \Delta + n^{-\frac{1}{\max\left\{2+\frac{\ell}{\alpha_g},2+\frac{s}{\alpha_e(\alpha_g\wedge 1)},\ell\right\}}}\log^2 n$, *for* $diss(\cdot,\cdot) = \mathcal{W}_1(\cdot,\cdot)$,

(b) $\mathbb{E}V(\mu, \nu, \hat{G}^n, \hat{E}^n) \lesssim \Delta + n^{-\frac{1}{2+\max\left\{\frac{\ell}{\alpha_g}, \frac{s}{\alpha_e(\alpha_g \wedge 1)}\right\}}} \log^2 n$, for $diss(\cdot, \cdot) = MMD_{\mathcal{K}}^2(\cdot, \cdot)$.

Furthermore, for the estimation problem (6), if $m \geq n \vee n^{\left(\max\left\{2+\frac{\ell}{\alpha_g}, 2+\frac{d_\mu}{\alpha_e(\alpha_g \wedge 1)}, \ell\right\}\right)^{-1}(\ell \vee 2)}$

(c) $\mathbb{E}V(\mu, \nu, \hat{G}^n, \hat{E}^n) \lesssim \Delta + n^{-\frac{1}{\max\left\{2+\frac{\ell}{\alpha_g}, 2+\frac{s}{\alpha_e(\alpha_g \wedge 1)}, \ell\right\}}} \log^2 n$, for $diss(\cdot, \cdot) = \mathcal{W}_1(\cdot, \cdot)$,

(d) $\mathbb{E}V(\mu, \nu, \hat{G}^{n,m}, \hat{E}^{n,m}) \lesssim \Delta + n^{-\frac{1}{2+\max\left\{\frac{\ell}{\alpha_g}, \frac{s}{\alpha_e(\alpha_g \wedge 1)}\right\}}} \log^2 n$, for $diss(\cdot, \cdot) = MMD_{\mathcal{K}}^2(\cdot, \cdot)$.

Before we proceed, we observe some key consequences of Theorem 8.

**Remark 9** (Number of Weights). We note that Theorem 8 suggests that one can choose the networks to have number of weights to be an exponent of $n$, which is smaller than 1. Moreover, this exponent only depends on the dimensions of the latent space and the intrinsic dimension of the data. Furthermore, for smooth models i.e. $\alpha_e$ and $\alpha_g$ are large, one can choose smaller networks that require less many parameters as opposed to non-smooth models as also observed in practice since easier problems require less complicated networks.

**Remark 10** (Rates for Lipschitz models). For all practical purposes, one can assume that the dimension of the latent space is at least 2. If the true models are Lipschitz, i.e. if $\alpha_e = \alpha_g = 1$, then we can conclude that $\ell = d_\mu$. Hence, for both models, we observe that the excess risk scales as $\tilde{\mathcal{O}}(n^{-\frac{1}{2+d_\mu}})$, barring the poly-log factors. This closely matches rates for the excess risks for GANs (Huang et al., 2022).

**Remark 11** (Inference for Data on Manifolds). We recall that we call a set $\mathcal{M}$ is $\tilde{d}$-regular w.r.t. the $\tilde{d}$-dimensional Hausdorff measure $\mathbb{H}^{\tilde{d}}$ if $\mathbb{H}(B_\varrho(x, r)) \asymp r^{\tilde{d}}$, for all $x \in \mathcal{M}$ (see Definition 6 of Weed and Bach (2019)). It is known (Mattila, 1999) that if $\mathcal{M}$ is $\tilde{d}$-regular, then the Minkowski dimension of $\mathcal{M}$ is $\tilde{d}$. Thus, when $\text{supp}(\mu)$ is $\tilde{d}$-regular, $d_\mu = \tilde{d}$. Since compact $\tilde{d}$-dimensional differentiable manifolds are $\tilde{d}$-regular (Proposition 9 of Weed and Bach (2019)), this implies that for when $\text{supp}(\mu)$ is a compact differentiable $\tilde{d}$-dimensional manifold, the error rates for the sample estimates scale as in Theorem 8, with $d_\mu$ replaced with $\tilde{d}$. A similar result holds when $\text{supp}(\mu)$ is a nonempty, compact convex set spanned by an affine space of dimension $\tilde{d}$; the relative boundary of a nonempty, compact convex set of dimension $\tilde{d} + 1$; or self-similar set with similarity dimension $\tilde{d}$.

## 5.3 RELATED WORK ON GANS

To contextualize our contributions, we conduct a qualitative comparison with existing GAN literature. Notably, Chen et al. (2020) expressed the generalization rates for GAN when the data is restricted to an affine subspace or has a mixture representation with smooth push-forward measures; while Dahal et al. (2022) derived the convergence rates under the Wasserstein-1 distance in terms of the manifold dimension. Both Liang (2021) and Schreuder et al. (2021) study the expected excess risk of GANs for smooth generator and discriminator function classes. Liu et al. (2021) studied the properties of Bidirectional GANs, expressing the rates in terms of the number of data points, where the exponents depend on the full data and latent space dimensions. It is important to note that both Dahal et al. (2022) and Liang (2021) assume that the densities of the target distribution (either w.r.t Hausdorff or the Lebesgue measure) are bounded and smooth. In comparison, we do not make any assumption of the existence of density (or its smoothness) for the target distribution and consider the practical case where the generator is realized through neural networks as opposed to smooth functions as done by Liang (2021) and Schreuder et al. (2021). Diverging from the hypotheses of Chen et al. (2020), we do not presuppose that the support of the target measure forms an affine subspace. Furthermore, the analysis by Liu et al. (2021) derives rates that depend on the dimension of the entire space and not the manifold dimension of the support of the data as done in this analysis. It is important to emphasize that Huang et al. (2022) arrived at a rate comparable to ours concerning WGANs (Arjovsky et al., 2017). While both studies share a common overarching approach in addressing the problem by bounding the error using an oracle inequality and managing individual terms, our method necessitates extra assumptions to guarantee the generative capability of WAEs, which does not apply to WGANs due to their simpler structure. Interestingly, our derived rates closely resemble those found in GAN literature. This suggests limited room for substantial

improvement. However, demonstrating minimaxity remains a significant challenge and a promising avenue for future research.

## 5.4 PROOF OVERVIEW

From Lemma 7, it is clear that the expected excess risk can be bounded by the misspecification error $\Delta_{\mathrm{miss}}$ and the generalization gap, $\|\hat\mu_n - \mu\|_{\mathcal{F}} + \lambda \sup_{E\in\mathcal{E}} |\widehat{\mathrm{diss}}(E_\sharp\hat\mu_n,\nu) - \mathrm{diss}(E_\sharp\mu,\nu)|$. To control $\Delta_{\mathrm{miss}}$, we first show that if the generator and encoders are chosen as $\mathcal{G} = \mathcal{RN}(W_g, L_g)$ and $\mathcal{E} = \mathcal{RN}(W_e, L_e)$, with $L_e \leq \alpha_0 \log(1/\epsilon_g)$, $L_g \leq \alpha_0 \log(1/\alpha_g)$, $W_e \leq \alpha_0 \epsilon_e^{-s/\alpha_e} \log(1/\epsilon_e)$ and $W_g \leq \alpha_0 \epsilon_g^{-\ell/\alpha_g} \log(1/\epsilon_g)$ then, $\Delta_{\mathrm{miss}} \lesssim \epsilon_g + \epsilon_e^{\alpha_g \wedge 1}$. On the other hand, we show that the generalization error is roughly $\sqrt{n^{-1} W_e L_e \log W_e \log n} + \sqrt{n^{-1}(W_e + W_g)(L_e + L_g) \log(W_e + W_g) \log n}$, with additional terms depending on the estimator. Thus, the bounds in Lemma 7, leads to a bound roughly,

$$\Delta + \epsilon_g + \epsilon_e^{\alpha_g \wedge 1} + \sqrt{n^{-1} W_e L_e \log W_e \log n} + \sqrt{n^{-1}(W_e + W_g)(L_e + L_g) \log(W_e + W_g) \log n}. \quad (9)$$

By the choice of the networks, we can upper bound the above as a function of $\epsilon_g$ and $\epsilon_e$ and then minimize the expression w.r.t. these two variables to arrive at the bounds of Theorem 8. Of course, the bound in (9) changes slightly based on the estimates and the dissimilarity measure. We refer the reader to the appendix, which contains the details of the proof.

## 5.5 IMPLICATIONS OF THE THEORETICAL RESULTS

Apart from finding the error rates for the excess risk for the WAE problem, in what follows, we also ensure a few desirable properties of the obtained estimates. For simplicity, we ignore the optimization error and set $\Delta_{\mathrm{opt}} = 0$.

**Encoding Guarantee** Suppose we fix $\lambda > 0$, then, it is clear from Theorem 8 that $\mathbb{E}\mathcal{W}_1(\hat E_\sharp\mu, \nu) \lesssim n^{-\frac{1}{\max\left\{2 + \frac{\ell}{\alpha_g}, 2 + \frac{s}{\alpha_e(\alpha_g\wedge 1)}, \ell\right\}}} \log^2 n$ and $\mathbb{E}\mathrm{MMD}_{\mathcal{K}}^2(\hat E_\sharp\mu, \nu) \lesssim n^{-\frac{1}{2 + \max\left\{\frac{\ell}{\alpha_g}, \frac{s}{\alpha_e(\alpha_g\wedge 1)}\right\}}} \log^2 n$. We can not only characterize the expected rate of convergence of $\hat E_\sharp\mu$ to $\nu$ but also can say that $\hat E_\sharp\mu$ converges in distribution to $\nu$, almost surely. This is formally stated in the following proposition.

**Proposition 12.** *Suppose that assumptions A1–3 hold. Then, for both the dissimilarity measures* $\mathcal{W}_1(\cdot,\cdot)$ *and* $MMD_{\mathcal{K}}^2(\cdot,\cdot)$ *and the estimates* (5) *and* (6), $\hat E_\sharp\mu \xrightarrow{d} \nu$, *almost surely.*

Therefore, if the number of data points is large, i.e., $n$ is large, then the estimated encoded distribution $\hat E_\sharp\mu$ will converge to the true target latent distribution $\nu$ almost surely, indicating that the latent distribution can be approximated through encoding with a high degree of accuracy.

**Decoding Guarantee** One can also show that $\mathbb{E} \int c(x, \hat G \circ \hat E(x)) d\mu(x) \leq \mathbb{E}V(\mu, \nu, \hat G, \hat E)$, for both the estimates in (5) and (6). For simplicity, if we let $c(x, y) = \|x - y\|_2^2$, then, it can easily seen that, $\mathbb{E}\|id(\cdot) - \hat G \circ \hat E(\cdot)\|_{\mathbb{L}_2(\mu)}^2 \to 0$ as $n \to \infty$, where $id(x) = x$ is the identity map from $\mathbb{R}^d \to \mathbb{R}^d$. Furthermore, it can be shown that, $\|id(\cdot) - \hat G \circ \hat E(\cdot)\|_{\mathbb{L}_2(\mu)}^2 \xrightarrow{a.s.} 0$ as stated in Corollary 13

**Proposition 13.** *Suppose that assumptions A1–3 hold. Then, for both the dissimilarity measures* $\mathcal{W}_1(\cdot,\cdot)$ *and* $MMD_{\mathcal{K}}^2(\cdot,\cdot)$ *and the estimates* (5) *and* (6), $\|id(\cdot) - \hat G \circ \hat E(\cdot)\|_{\mathbb{L}_2(\mu)}^2 \xrightarrow{a.s.} 0$.

Proposition 13 guarantees that the generator is able to map back the encoded points to the original data if a sufficiently large amount of data is available. In other words, if one has access to a large number of samples from the data distribution, then the generator is able to learn a mapping, from the encoded points to the original data, that is accurate enough to be useful.

**Data Generation Guarantees** A key interest in this theoretical exploration is whether one can guarantee that one can generate samples from the unknown target distribution $\mu$, through the generator, i.e. whether $\hat G_\sharp\nu$ is close enough to $\mu$ in some sense. However, one requires some additional assumptions (Chakrabarty & Das, 2021; Tang & Yang, 2021) on $\hat G$ or the nature of convergence

of $\hat{E}_\sharp\mu$ to $\nu$ to ensure this. We present the corresponding results subsequently as follows. Before proceeding, we recall the definition of Total Variation (TV) distance between two measures $\gamma_1$ and $\gamma_2$, defined on $\Omega$, as, $TV(\gamma_1, \gamma_2) = \sup_{B \in \mathcal{B}(\Omega)} |\gamma_1(B) - \gamma_2(B)|$, where, $\mathcal{B}(\Omega)$ denotes the Borel $\sigma$-algebra on $\Omega$.

**Theorem 14.** *Suppose that assumptions A1–3 hold and $TV(\hat{E}_\sharp\mu, \nu) \to 0$, almost surely. Then, $\hat{G}_\sharp\nu \xrightarrow{d} \mu$, almost surely.*

We note that convergence in TV is a much stronger assumption than convergence in $\mathcal{W}_1$ or MMD in the sense that TV convergence implies weak convergence but not the other way around.

Another way to ensure that $\hat{G}_\sharp\nu$ converges to $\mu$ is to put some sort of regularity on the generator estimates. Tang & Yang (2021) imposed a Lipschitz assumption to ensure this, but one can also work with something weaker, such as uniform equicontinuity of the generators. Recall that we say a family of functions, $\mathcal{F}$ is uniformly equicontinuous if, for any $f \in \mathcal{F}$ and for all $\epsilon > 0$, there exists a $\delta > 0$ such that, $|f(x) - f(y)| \le \epsilon$, whenever, $\|x - y\| \le \delta$.

**Theorem 15.** *Suppose that assumptions A1–3 hold and let the family of estimated generators $\{\hat{G}^n\}_{n \in \mathbb{N}}$ be uniformly equicontinuous, almost surely. Then, $\hat{G}^n_\sharp\nu \xrightarrow{d} \mu$, almost surely.*

**Uniformly Lipschitz Generators**  Suppose that diss$(\cdot, \cdot) = \mathcal{W}_1(\cdot, \cdot)$. If one assumes that the estimated generators are uniformly Lipschitz, then, one can say that $\mathcal{W}_1(\hat{G}_\sharp\nu, \mu)$ is upper bounded by $V(\mu, \nu, \hat{G}, \hat{E})$, disregarding some constants. Thus, the same rate of convergence as in Theorem 8 holds for uniformly Lipschitz generator. We state this result formally as a corollary as follows.

**Corollary 16.** *Let diss$(\cdot, \cdot) = \mathcal{W}_1(\cdot, \cdot)$ and suppose that the assumptions of Theorem 8 are satisfied and $s > d_\mu$. Also let $\sup_{n \in \mathbb{N}} \|\hat{G}^n\|_{Lip}, \sup_{m,n \in \mathbb{N}} \|\hat{G}^{n,m}\|_{Lip} \le L$, almost surely, for some $L > 0$. $\mathcal{W}_1(\hat{G}_\sharp\nu, \mu) \lesssim V(\mu, \nu, \hat{G}, \hat{E})$ for both estimators (5) and (6).*

It is important to note that although assumptions A1–3 do not directly guarantee either of these two conditions, it is reasonable to expect the assumptions made in Theorems 14 and 15 to hold in practice. This is because regularization techniques are commonly used to ensure the learned networks $\hat{E}$ and $\hat{G}$ are sufficiently well-behaved. These techniques can impose various constraints, such as weight decay or dropout, that encourage the networks to have desirable properties, such as smoothness or sparsity. Therefore, while the assumptions made in the theorems cannot be directly ensured by A1–3, they are likely to hold in practice with appropriate regularization techniques applied to the network training. It would be a key step in furthering our understanding to develop a similar error analysis for such regularized networks and we leave this as a promising direction for future research.

## 6 DISCUSSIONS AND CONCLUSION

In this paper, we developed a framework to analyze error rates for learning unknown distributions using Wasserstein Autoencoders, especially when data points exhibit an intrinsically low-dimensional structure in the representative high-dimensional feature space. We characterized this low dimensionality with the so-called Minkowski dimension of the support of the target distribution. We developed an oracle inequality to characterize excess risk in terms of misspecification, generalization, and optimization errors for the problem. The excess risk bounds are obtained by balancing model-misspecification and stochastic errors to find proper network architectures in terms of the number of samples that achieve this tradeoff. Our framework allows us to analyze the accuracy of encoding and decoding guarantees, i.e., how well the encoded distribution approximates the target latent distribution, and how well the generator maps back the latent codes close to the original data points. Furthermore, with additional regularity assumptions, we establish that the approximating push-forward measure can effectively approximate the target distribution.

While our findings provide valuable insights into the theoretical characteristics of Wasserstein Autoencoders (WAEs), it's crucial to acknowledge that achieving accurate estimates of the overall error in practical applications necessitates the consideration of an optimization error term. However, the precise estimation of this term poses a significant challenge due to the non-convex and intricate nature of the optimization process. Importantly, our error analysis remains independent of this optimization error and can seamlessly integrate with analyses involving such optimization complexities.

ACKNOWLEDGMENT

We gratefully acknowledge the support of the NSF and the Simons Foundation for the Collaboration on the Theoretical Foundations of Deep Learning through awards DMS-2031883 and #814639 and the NSF's support of FODSI through grant DMS-2023505.

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

# Appendix

CONTENTS

## A  ADDITIONAL NOTATIONS

For function classes $\mathcal{F}_1$ and $\mathcal{F}_2$, $\mathcal{F}_1 \circ \mathcal{F}_2 = \{f_1 \circ f_2 : f_1 \in \mathcal{F}_1,\ f_2 \in \mathcal{F}_2\}$.

**Definition 17** (Covering and Packing Numbers). For a metric space $(\mathcal{S}, \varrho)$, the $\epsilon$-covering number w.r.t. $\varrho$ is defined as: $\mathcal{N}(\epsilon; \mathcal{S}, \varrho) = \inf\{n \in \mathbb{N} : \exists x_1, \dots x_n \text{ such that } \cup_{i=1}^n B_\varrho(x_i, \epsilon) \supseteq \mathcal{S}\}$. A minimal $\epsilon$ cover of $\mathcal{S}$ is denoted as $\mathcal{C}(\epsilon; \mathcal{S}, \varrho)$. Similarly, the $\epsilon$-packing number is defined as: $\mathcal{M}(\epsilon; \mathcal{S}, \varrho) = \sup\{m \in \mathbb{N} : \exists x_1, \dots x_m \in \mathcal{S} \text{ such that } \varrho(x_i, x_j) \geq \epsilon, \text{ for all } i \neq j\}$.

## B  PROOF OF THE MAIN RESULT (THEOREM 8)

### B.1  MISSPECIFICATION ERROR

We begin with a theoretical result to approximate any function on a low-dimensional structure using a ReLU network with sufficiently large depth and width. Let $f$ belong to the space $\mathcal{H}^\beta(\mathbb{R}^d, \mathbb{R}, C)$, with $C > 0$, and let $\gamma$ be a measure on $\mathbb{R}^d$. For notational simplicity, let $\mathcal{M} = \text{supp}(\gamma)$. Then, for any $\epsilon > 0$ and $s > d_\gamma$, we prove that there exists a ReLU network, denoted by $\hat{f}$, with a depth of at most $\mathcal{O}(\log(1/\epsilon))$, and number of weights not exceeding $\mathcal{O}(\epsilon^{-s/\beta} \log(1/\epsilon))$, and bounded weights. This network satisfies the condition $\|f - \hat{f}\|_{\mathbb{L}_\infty(\mathcal{M})} \leq \epsilon$. A similar result with bounded depth but unbounded weights was derived by Nakada & Imaizumi (2020).

**Theorem 18.** *Let $f$ be an element of $\mathcal{H}^\beta(\mathbb{R}^d, \mathbb{R}, C)$, where $C > 0$. Then, for any $s > d_\gamma$, there exists constants $\epsilon_0$ (which may depend on $\gamma$) and $\alpha$, (which may depend on $\beta$, $d$, and $C$), such that if $\epsilon \in (0, \epsilon_0]$, a ReLU network $\hat{f}$ can be constructed with $\mathcal{L}(\hat{f}) \leq \alpha \log(1/\epsilon)$ and $\mathcal{W}(\hat{f}) \leq \alpha \log(1/\epsilon)\epsilon^{-s/\beta}$, satisfying the condition, $\|f - \hat{f}\|_{\mathbb{L}_\infty(\mathcal{M})} \leq \epsilon$.*

Applying the above theorem, one can control $\Delta_{\text{miss}}$, when the network size is large enough. Under assumptions A1–3, we derive the following bound on the misspecification error. It is important to note that none of the network dimensions depend on the dimensionality of the entire data space, i.e. $d$.

**Lemma 19.** *Suppose assumptions A1–3 hold and let, $\text{diss}(\cdot, \cdot) \equiv \mathcal{W}_1(\cdot, \cdot)$ or $MMD_\mathcal{K}^2(\cdot, \cdot)$. Also, let $s > d_\mu$. Then, we can find positive constants $\epsilon_0$, $\alpha_0$ and $R$, that might depend on $d, \ell, \tilde{G}$ and $\tilde{E}$, such that if $0 < \epsilon_g, \epsilon_e \leq \epsilon_0$ and $\mathcal{G} = \mathcal{RN}(W_g, L_g, R)$ and $\mathcal{E} = \mathcal{RN}(W_e, L_e, R)$, with*

$$L_e \leq \alpha_0 \log(1/\epsilon_g),\ L_g \leq \alpha_0 \log(1/\alpha_g),\ W_e \leq \alpha_0 \epsilon_e^{-s/\alpha_e} \log(1/\epsilon_e) \text{ and } W_g \leq \alpha_0 \epsilon_g^{-\ell/\alpha_g} \log(1/\epsilon_g)$$

*then, $\Delta_{\text{miss}} \lesssim \epsilon_g + \epsilon_e^{\alpha_g \wedge 1}$.*

### B.2 BOUNDING THE GENERALIZATION GAP

Let $f : \mathbb{R}^d \to \mathbb{R}^{d'}$ and $\{X_i\}_{i \in [n]} \subset \mathbb{R}^d$. We define $f_{|X_{1:n}}$ as $[f(X_1) : \cdots : f(X_n)] \in \mathbb{R}^{d' \times n}$. For a function class $\mathcal{F}$, we define

$$\mathcal{F}_{|X_{1:n}} = \{f_{|X_{1:n}} : f \in \mathcal{F}\} \subseteq \mathbb{R}^{d' \times n}.$$

The covering number of $\mathcal{F}_{|X_{1:n}}$ with respect to the $\ell_\infty$-norm is denoted by $\mathcal{N}(\epsilon; \mathcal{F}_{|X_{1:n}}, \ell_\infty)$. The result extends the seminal works of Bartlett et al. (2019) to determine the metric entropy of deep learners with multivariate outputs.

**Lemma 20.** *Suppose that $n \geq 6$ and $\mathcal{F}$ are a class neural network with depth at most $L$ and number of weights at most $W$. Furthermore, the activation functions are piece-wise polynomial activation with the number of pieces and degree at most $k \in \mathbb{N}$. Then, there is a constant $\theta$ (that might depend on $d$ and $d'$), such that, if $n \geq \theta(W + 6d' + 2d'L)(L + 3)(\log(W + 6d' + 2d'L) + L + 3)$,*

$$\log \mathcal{N}(\epsilon; \mathcal{F}_{|X_{1:n}}, \ell_\infty) \lesssim (W + 6d' + 2d'L)(L + 3)(\log(W + 6d' + 2d'L) + L + 3) \log\left(\frac{nd'}{\epsilon}\right),$$

*where $d'$ is the output dimension of the networks in $\mathcal{F}$.*

We can use the result above to provide bounds on the metric entropies of the function classes described in Lemma 7. This bound is a function of the number of samples and the size of the network classes $\mathcal{G}$ and $\mathcal{E}$.

**Corollary 21.** *Suppose that $\mathcal{W}(\mathcal{E}) \leq W_e$, $\mathcal{L}(\mathcal{E}) \leq L_e$, $\mathcal{W}(\mathcal{G}) \leq W_g$ and $\mathcal{L}(\mathcal{G}) \leq L_g$, with $L_e, L_g \geq 3$, $W_e \geq 6\ell + 2\ell L_e$ and $W_g \geq 6d + 2dL_g$. Then, there is a constant $\xi_1$, such that if $n \geq \xi_1(W_e + W_g)(L_e + L_g)(\log(W_e + W_g) + L_e + L_g)$,*

$$\log \mathcal{N}\left(\epsilon; \mathcal{E}_{|X_{1:n}}, \ell_\infty\right) \lesssim W_e L_e (\log W_e + L_e) \log\left(\frac{n\ell}{\epsilon}\right),$$

$$\log \mathcal{N}\left(\epsilon; (\mathcal{G} \circ \mathcal{E})_{|X_{1:n}}, \ell_\infty\right) \lesssim (W_e + W_g)(L_e + L_g)(\log(W_e + W_g) + L_e + L_g) \log\left(\frac{nd}{\epsilon}\right).$$

Using Corollary 21, the following lemma provides a bound on the distance between the empirical and target distributions w.r.t. the IPM based on $\mathcal{F}$.

**Lemma 22.** *Suppose $\mathcal{R}(\mathcal{G}) \lesssim 1$ and $\mathcal{F} = \{f(x) = c(x, G \circ E(x)) : G \in \mathcal{G}, E \in \mathcal{E}\}$. Furthermore, let, $\mathcal{L}(\mathcal{E}) \leq L_e$, $\mathcal{W}(\mathcal{G}) \leq W_g$ and $\mathcal{L}(\mathcal{G}) \leq L_g$, with $L_e, L_g \geq 3$, $W_e \geq 6\ell + 2\ell L_e$ and $W_g \geq 6d + 2dL_g$. Then, there is a constant $\xi_2$, such that if $n \geq \xi_2(W_e + W_g)(L_e + L_g)(\log(W_e + W_g) + L_e + L_g)$*

$$\mathbb{E}\|\hat{\mu}_n - \mu\|_{\mathcal{F}} \lesssim n^{-1/2}\left((W_e + W_g)(L_e + L_g)\left(\log(W_e + W_g) + L_e + L_g\right)\log(nd)\right)^{1/2}.$$

To control the fourth terms in (7) and (8), we first consider the case when $\mathrm{diss}(\cdot, \cdot)$ is the $\mathcal{W}_1$-distance. Lemma 23 controls this uniform concentration via the size of the networks in $\mathcal{E}$ and the sample size $n$.

**Lemma 23.** *Let $\hat{\mu}_n = \frac{1}{n}\sum_{i=1}^n \delta_{X_i}$ and $\mathcal{E} = \mathcal{RN}(L_e, W_e)$. Then,*

$$\sup_{E \in \mathcal{E}} |\mathcal{W}_1(E_\sharp \mu_n, \nu) - \mathcal{W}_1(E_\sharp \mu, \nu)| \lesssim \left(n^{-1/\ell} \vee n^{-1/2}\log n\right) + \sqrt{\frac{W_e L_e(\log W_e + L_e)\log(n\ell)}{n}}.$$

*Furthermore,*

$$\sup_{E \in \mathcal{E}} |\mathcal{W}_1(E_\sharp \hat{\mu}_n, \hat{\nu}_m) - \mathcal{W}_1(E_\sharp \mu, \nu)| \lesssim \left(n^{-1/\ell} \vee n^{-1/2}\log n\right) + \left(m^{-1/\ell} \vee m^{-1/2}\log m\right)$$

$$+ \sqrt{\frac{W_e L_e(\log W_e + L_e)\log(n\ell)}{n}}.$$

Before deriving the corresponding uniform concentration bounds for $|\widehat{\mathrm{MMD}}_{\mathcal{K}}^2(E_\sharp \hat{\mu}_n, \nu) - \mathrm{MMD}_{\mathcal{K}}^2(E_\sharp \hat{\mu}_n, \nu)|$ or $|\widehat{\mathrm{MMD}}_{\mathcal{K}}^2(E_\sharp \hat{\mu}_n, \hat{\nu}_m) - \mathrm{MMD}_{\mathcal{K}}^2(E_\sharp \hat{\mu}_n, \nu)|$, we recall the definition of

Rademacher complexity (Bartlett & Mendelson, 2002). For any real-valued function class $\mathcal{F}$ and data points $X_{1:n} = \{X_1, \ldots, X_n\}$, the empirical Rademacher complexity is defined as:

$$\mathcal{R}(\mathcal{F}, X_{1:n}) = \frac{1}{n} \mathbb{E}_{\boldsymbol{\sigma}} \sup_{f \in \mathcal{F}} \sum_{i=1}^{n} \sigma_i f(X_i),$$

where $\sigma_i$'s are i.i.d. Rademacher random variables taking values in $\{-1, +1\}$, with equal probability. In the following lemma, we derive a bound on the Rademacher complexity of the class of functions in the unit ball w.r.t. the $\mathbb{H}_{\mathcal{K}}$-norm composed with $\mathcal{E}$. This lemma plays a key role in the proof of Lemma 24. The proof crucially uses the results by Rudelson & Vershynin (2013).

**Lemma 24.** *Suppose assumption A2 holds and let, $\mathcal{L}(\mathcal{E}) \leq L_e$ and $\mathcal{L}(\mathcal{G}) \leq L_g$, with $L_e \geq 3$, $W_e \geq 2\ell(3 + L_e)$. Also suppose that, $\Phi = \{\phi \in \mathbb{H}_{\mathcal{K}} : \|\phi\|_{\mathbb{H}_{\mathcal{K}}} \leq 1\}$, then,*

$$\mathcal{R}((\Phi \circ \mathcal{E}), X_{1:n}) \lesssim \sqrt{\frac{W_e L_e (\log W_e + L_e) \log(n\ell)}{n}}.$$

Using Lemma 24, we bound the fourth term in (7) and (8) for $\mathrm{diss}(\cdot, \cdot) = \mathrm{MMD}_{\mathcal{K}}^2(\cdot, \cdot)$, in Lemma 25.

**Lemma 25.** *Under assumption A2, the following holds:*

*(a)* $\mathbb{E} \sup_{E \in \mathcal{E}} \left| \widehat{MMD}_{\mathcal{K}}^2(E_\sharp \hat{\mu}_n, \nu) - MMD_{\mathcal{K}}^2(E_\sharp \mu, \nu) \right| \lesssim \sqrt{\frac{W_e L_e \log W_e \log(n\ell)}{n}},$

*(b)* $\mathbb{E} \sup_{E \in \mathcal{E}} \left| \widehat{MMD}_{\mathcal{K}}^2(E_\sharp \hat{\mu}_n, \hat{\nu}_m) - MMD_{\mathcal{K}}^2(E_\sharp \mu, \nu) \right| \lesssim \sqrt{\frac{W_e L_e (\log W_e + L_e) \log(n\ell)}{n}} + \frac{1}{\sqrt{m}}.$

### B.3 PROOF OF THEOREM 8

**Theorem 8.** *Suppose that assumptions A1–3 hold and $\Delta_{opt} \leq \Delta$ for some fixed non-negative threshold $\Delta$. Furthermore, suppose that $s > d_\mu$. Then we can find $n_0 \in \mathbb{N}$ and $\beta > 0$, that might depend on $d, \ell, \alpha_g, \alpha_e, \tilde{G}$ and $\tilde{E}$, such that if $n \geq n_0$, we can choose $\mathcal{G} = \mathcal{RN}(L_g, W_g)$ and $\mathcal{E} = \mathcal{RN}(L_e, W_e)$, with, $L_e \leq \beta \log n$, $W_e \leq \beta n^{\frac{s}{2\alpha_e + s}} \log n$, $L_g \leq \beta \log n$ and $W_g \leq \beta n^{\frac{\ell}{\alpha_e(\alpha_g \wedge 1) + \ell}} \log n$, then, for the estimation problem (5),*

*(a)* $\mathbb{E} V(\mu, \nu, \hat{G}^n, \hat{E}^n) \lesssim \Delta + n^{-\frac{1}{\max\left\{2 + \frac{\ell}{\alpha_g}, 2 + \frac{s}{\alpha_e(\alpha_g \wedge 1)}, \ell\right\}}} \log^2 n$, *for* $\mathrm{diss}(\cdot, \cdot) = \mathcal{W}_1(\cdot, \cdot)$,

*(b)* $\mathbb{E} V(\mu, \nu, \hat{G}^n, \hat{E}^n) \lesssim \Delta + n^{-\frac{1}{2 + \max\left\{\frac{\ell}{\alpha_g}, \frac{s}{\alpha_e(\alpha_g \wedge 1)}\right\}}} \log^2 n$, *for* $\mathrm{diss}(\cdot, \cdot) = \mathrm{MMD}_{\mathcal{K}}^2(\cdot, \cdot)$.

*Furthermore, for the estimation problem (6), if $m \geq n \vee n^{\left(\max\left\{2 + \frac{\ell}{\alpha_g}, 2 + \frac{d_\mu}{\alpha_e(\alpha_g \wedge 1)}, \ell\right\}\right)^{-1}(\ell \vee 2)}$*

*(c)* $\mathbb{E} V(\mu, \nu, \hat{G}^n, \hat{E}^n) \lesssim \Delta + n^{-\frac{1}{\max\left\{2 + \frac{\ell}{\alpha_g}, 2 + \frac{s}{\alpha_e(\alpha_g \wedge 1)}, \ell\right\}}} \log^2 n$, *for* $\mathrm{diss}(\cdot, \cdot) = \mathcal{W}_1(\cdot, \cdot)$,

*(d)* $\mathbb{E} V(\mu, \nu, \hat{G}^{n,m}, \hat{E}^{n,m}) \lesssim \Delta + n^{-\frac{1}{2 + \max\left\{\frac{\ell}{\alpha_g}, \frac{s}{\alpha_e(\alpha_g \wedge 1)}\right\}}} \log^2 n$, *for* $\mathrm{diss}(\cdot, \cdot) = \mathrm{MMD}_{\mathcal{K}}^2(\cdot, \cdot)$.

*Proof.* **Proof of part (a)** From Lemmas 7, 22 and 23, we get,

$\mathbb{E} V(\mu, \nu, \hat{G}^n, \hat{E}^n)$

$\lesssim \Delta + \epsilon_g + \epsilon_e^{\alpha_g \wedge 1} + \sqrt{\frac{W_e L_e \log W_e \log n}{n}} + \sqrt{\frac{(W_e + W_g)(L_e + L_g) \log(W_e + W_g) \log n}{n}} + \left(n^{-1/\ell} \vee n^{-1/2}\right)$

$\lesssim \Delta + \epsilon_g + \epsilon_e^{\alpha_g \wedge 1} + \left(\log\left(\frac{1}{\epsilon_e \wedge \epsilon_g}\right)\right)^{3/2} \left(\sqrt{\frac{\epsilon_e^{-s/\alpha_e} \log n}{n}} + \sqrt{\frac{(\epsilon_e^{-s/\alpha_e} + \epsilon_g^{-\ell/\alpha_g}) \log n}{n}}\right) + \left(n^{-1/\ell} \vee n^{-1/2}\right)$

$\lesssim \Delta + \epsilon_g + \epsilon_e^{\alpha_g \wedge 1} + \left(\log\left(\frac{1}{\epsilon_e \wedge \epsilon_g}\right)\right)^{3/2} \left(\sqrt{\frac{\epsilon_e^{-s/\alpha_e} \log n}{n}} + \sqrt{\frac{\epsilon_g^{-\ell/\alpha_g} \log n}{n}}\right) + \left(n^{-1/\ell} \vee n^{-1/2}\right)$

We choose, $\epsilon_g \asymp n^{-\frac{1}{2+\frac{\ell}{\alpha_g}}}$ and $\epsilon_e \asymp n^{-\frac{1}{2(\alpha_g \wedge 1)+\frac{s}{\alpha_e}}}$. This makes,

$$\mathbb{E}V(\mu,\nu,\hat{G}^n,\hat{E}^n) \lesssim \Delta + \log^2 n \times n^{-\frac{1}{\max\left\{2+\frac{\ell}{\alpha_g},2+\frac{s}{\alpha_e(\alpha_g \wedge 1)}\right\}}} + n^{-1/\ell}.$$

**Proof of part (b)** From Lemmas 7, 22 and 25, we get,

$$\mathbb{E}V(\mu,\nu,\hat{G}^n,\hat{E}^n)$$

$$\lesssim \Delta + \epsilon_g + \epsilon_e^{\alpha_g \wedge 1} + \sqrt{\frac{W_e L_e \log W_e \log n}{n}} + \sqrt{\frac{(W_e + W_g)(L_e + L_g)\log(W_e + W_g)\log n}{n}}$$

$$\lesssim \Delta + \epsilon_g + \epsilon_e^{\alpha_g \wedge 1} + \left(\log\left(\frac{1}{\epsilon_e \wedge \epsilon_g}\right)\right)^{3/2}\left(\sqrt{\frac{\epsilon_e^{-s/\alpha_e}\log n}{n}} + \sqrt{\frac{(\epsilon_e^{-s/\alpha_e}+\epsilon_g^{-\ell/\alpha_g})\log n}{n}}\right)$$

$$\lesssim \Delta + \epsilon_g + \epsilon_e^{\alpha_g \wedge 1} + \left(\log\left(\frac{1}{\epsilon_e \wedge \epsilon_g}\right)\right)^{3/2}\left(\sqrt{\frac{\epsilon_e^{-s/\alpha_e}\log n}{n}} + \sqrt{\frac{\epsilon_g^{-\ell/\alpha_g}\log n}{n}}\right)$$

We choose, $\epsilon_g \asymp n^{-\frac{1}{2+\frac{\ell}{\alpha_g}}}$ and $\epsilon_e \asymp n^{-\frac{1}{2(\alpha_g \wedge 1)+\frac{s}{\alpha_e}}}$. This makes,

$$\mathbb{E}V(\mu,\nu,\hat{G}^n,\hat{E}^n) \lesssim \Delta + \log^2 n \times n^{-\frac{1}{\max\left\{2+\frac{\ell}{\alpha_g},2+\frac{s}{\alpha_e(\alpha_g \wedge 1)}\right\}}}.$$

**Proof of part (c)**

Again from Lemmas 7, 22 and 23, we get,

$$\mathbb{E}V(\mu,\nu,\hat{G}^n,\hat{E}^n) \lesssim \Delta + \epsilon_g + \epsilon_e^{\alpha_g \wedge 1} + \left(\log\left(\frac{1}{\epsilon_e \wedge \epsilon_g}\right)\right)^{3/2}\left(\sqrt{\frac{\epsilon_e^{-s/\alpha_e}\log n}{n}} + \sqrt{\frac{\epsilon_g^{-\ell/\alpha_g}\log n}{n}}\right)$$

$$+ \left(n^{-1/\ell} \vee n^{-1/2}\right) + \left(m^{-1/\ell} \vee m^{-1/2}\right)$$

Choosing $\epsilon_g \asymp n^{-\frac{1}{2+\frac{\ell}{\alpha_g}}}, \epsilon_e \asymp n^{-\frac{1}{2(\alpha_g \wedge 1)+\frac{s}{\alpha_e}}}$ and $m$ as in the theorem statement gives us the desired result.

**Proof of part (d)** Similarly, from Lemmas 7, 22 and 25, we get,

$$\mathbb{E}V(\mu,\nu,\hat{G}^n,\hat{E}^n) \lesssim \Delta + \epsilon_g + \epsilon_e^{\alpha_g \wedge 1} + \left(\log\left(\frac{1}{\epsilon_e \wedge \epsilon_g}\right)\right)^{3/2}\left(\sqrt{\frac{\epsilon_e^{-s/\alpha_e}\log n}{n}} + \sqrt{\frac{\epsilon_g^{-\ell/\alpha_g}\log n}{n}}\right) + \frac{1}{\sqrt{m}}$$

Choosing $\epsilon_g \asymp n^{-\frac{1}{2+\frac{\ell}{\alpha_g}}}, \epsilon_e \asymp n^{-\frac{1}{2(\alpha_g \wedge 1)+\frac{s}{\alpha_e}}}$ and $m$ as in the theorem statement gives us the desired result. $\square$

## C   DETAILED PROOFS

### C.1   PROOFS FROM SECTION 4

**Lemma 5.** *Let,* $f \in \mathcal{H}^\gamma\left(\mathcal{A},[0,1]^{d_2},C\right)$, *with* $\mathcal{A} \subseteq [0,1]^{d_1}$. *Then,* $\overline{dim}_M\left(f\left(\mathcal{A}\right)\right) \leq \overline{dim}_M(\mathcal{A})/(\gamma \wedge 1)$.

*Proof.* Then, for any $x,y \in \mathcal{A}$, $\|f(x)-f(y)\|_\infty \leq \|f(x)-f(y)\|_2 \leq L\|x-y\|_2^{\gamma \wedge 1} \leq Ld_1^{(\gamma \wedge 1)/2}\|x-y\|_\infty^{\gamma \wedge 1}$. Thus, $\mathcal{N}\left(\epsilon;f\left(\mathcal{A}\right),\|\cdot\|_\infty\right) \leq \mathcal{N}\left(\frac{1}{\sqrt{d_1}}(\epsilon/L)^{(\gamma \wedge 1)^{-1}};\mathcal{A},\|\cdot\|_\infty\right)$.

$$\dim_M\left(f\left(\mathcal{A}\right)\right) = \lim_{\epsilon \to 0}\frac{\log \mathcal{N}\left(\epsilon;f\left(\mathcal{A}\right),\|\cdot\|_\infty\right)}{\log(1/\epsilon)} \leq \lim_{\epsilon \to 0}\frac{\log \mathcal{N}\left(\frac{1}{\sqrt{d_1}}(\epsilon/L)^{(\gamma \wedge 1)^{-1}};\mathcal{A},\|\cdot\|_\infty\right)}{\log(1/\epsilon)} \leq \frac{\overline{\dim}_M(\mathcal{A})}{\gamma \wedge 1}.$$

$\square$

## C.2 PROOFS FROM SECTION 5.1

### C.2.1 PROOF OF PROPOSITION 6

**Proposition 6.** *Under assumption A1, the following holds:* $(a)\, \tilde{G}_\sharp \nu = \mu$, $(b)\, (\tilde{E} \circ \tilde{G})_\sharp \nu = \nu$, $(c)\, V(\mu, \nu, \tilde{G}, \tilde{E}) = 0$.

*Proof.* (a) Let $f : \mathcal{Z} \to \mathbb{R}$ be any bounded continuous function. Then,

$$\int f(x)d(\tilde{G}_\sharp \nu)(x) = \int f(\tilde{G}(z))d\nu(z)$$

$$= \int f(\tilde{G}(\tilde{E}(x)))d\mu(x) \tag{10}$$

$$= \int f(x)d\mu(x) \tag{11}$$

Hence, $\tilde{G}_\sharp \nu = \mu$. Here both (10) and (11) follows from A1.

(b) Let $f : \mathcal{X} \to \mathbb{R}$ be any bounded continuous function. Then,

$$\int f(x)d\left((\tilde{E} \circ \tilde{G})_\sharp \nu\right) = \int f(E(G(z)))d\nu(z)$$

$$= \int f(E(x))d\mu(x) \tag{12}$$

$$= \int f(z)d\nu(z). \tag{13}$$

Here, (12) follows from part (a) and (13) follows from A1.

(c) To prove part (c), We note that, $\mathcal{W}_1(E_\sharp \mu, \nu), \mathrm{MMD}^2_{\mathcal{K}}(E_\sharp \mu, \nu) = 0$ and $(\tilde{G} \circ \tilde{E})(\cdot) = id(\cdot)$, a.e. $[\mu]$.

$\square$

### C.2.2 PROOF OF LEMMA 7

**Lemma 7** (Oracle Inequality). *Suppose that,* $\mathcal{F} = \{f(x) = c(x, G \circ E(x)) : G \in \mathcal{G}, E \in \mathcal{E}\}$. *Then the following hold:*

$$V(\mu, \nu, \hat{G}^n, \hat{E}^n) \leq \Delta_{miss} + \Delta_{opt} + 2\|\hat{\mu}_n - \mu\|_{\mathcal{F}} + 2\lambda \sup_{E \in \mathcal{E}} |\widehat{diss}(E_\sharp \hat{\mu}_n, \nu) - diss(E_\sharp \mu, \nu)|. \tag{7}$$

$$V(\mu, \nu, \hat{G}^{n,m}, \hat{E}^{n,m}) \leq \Delta_{miss} + \Delta_{opt} + 2\|\hat{\mu}_n - \mu\|_{\mathcal{F}} + 2\lambda \sup_{E \in \mathcal{E}} |\widehat{diss}(E_\sharp \hat{\mu}_n, \hat{\nu}_m) - diss(E_\sharp \mu, \nu)|. \tag{8}$$

*Here,* $\Delta_{miss} = \inf_{G \in \mathcal{G}, E \in \mathcal{E}} V(\mu, \nu, G, E)$ *denotes the misspecification error for the problem.*

*Proof.* To prove the first inequality, we observe that, $V(\hat{\mu}_n, \nu, \hat{G}^n, \hat{E}^n) \leq V(\hat{\mu}_n, \nu, G, E) + \Delta_{\mathrm{opt}}$, for any $G \in \mathcal{G}$ and $E \in \mathcal{E}$. Thus,

$$V(\mu, \nu, \hat{G}^n, \hat{E}^n)$$
$$= V(\mu, \nu, G, E) + \left(V(\mu, \nu, \hat{G}^n, \hat{E}^n) - V(\hat{\mu}_n, \nu, \hat{G}^n, \hat{E}^n)\right) + \left(V(\hat{\mu}_n, \nu, \hat{G}^n, \hat{E}^n) - V(\mu, \nu G, E)\right)$$
$$\leq V(\mu, \nu, G, E) + \left(V(\mu, \nu, \hat{G}^n, \hat{E}^n) - V(\hat{\mu}_n, \nu, \hat{G}^n, \hat{E}^n)\right) + (V(\hat{\mu}_n, \nu, G, E) - V(\mu, \nu, G, E)) + \Delta_{\mathrm{opt}}$$
$$\leq \Delta_{\mathrm{opt}} + V(\mu, \nu, G, E) + 2 \sup_{G \in \mathcal{G}, E \in \mathcal{E}} |V(\hat{\mu}_n, \nu, G, E) - V(\mu, \nu, G, E)|$$
$$= \Delta_{\mathrm{opt}} + V(\mu, \nu, G, E)$$
$$\quad + 2 \sup_{G \in \mathcal{G}, E \in \mathcal{E}} \left|\int c(x, G \circ E(x))d\hat{\mu}_n(x) + \lambda\widehat{\mathrm{diss}}(E_\sharp \hat{\mu}_n, \nu) - \int c(x, G \circ E(x))d\mu(x) - \lambda\mathrm{diss}(E_\sharp \mu, \nu)\right|$$

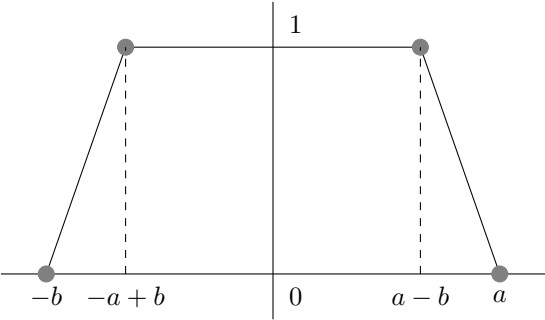

Figure 2: Plot of $\xi_{a,b}(\cdot)$

$$\leq \Delta_{\text{opt}} + V(\mu, \nu, G, E) + 2\|\hat{\mu}_n - \mu\|_{\mathcal{F}} + 2\lambda \sup_{E \in \mathcal{E}} |\widehat{\text{diss}}(E_\sharp \hat{\mu}_n, \nu) - \text{diss}(E_\sharp \mu, \nu)|.$$

Taking infimum on $G$ and $E$, we get inequality (7). Inequality (8) follows from a similar derivation. $\qquad\square$

### C.3 Proofs from Section B.1

#### C.3.1 Proof of Theorem 18

Fix $\epsilon > 0$. For $s > d_\mu$, we can say that $\mathcal{N}(\epsilon; \mathcal{M}, \ell_\infty) \leq C\epsilon^{-s}$, for all $\epsilon > 0$. Let $K = \lceil \frac{1}{2\epsilon} \rceil$. For any $\boldsymbol{i} \in [K]^d$, let $\boldsymbol{\theta^i} = (\epsilon + 2(i_1 - 1)\epsilon, \ldots, \epsilon + 2(i_d - 1)\epsilon)$. We also let, $\mathcal{P}_\epsilon = \{B_{\ell_\infty}(\boldsymbol{\theta^i}, \epsilon) : \boldsymbol{i} \in [K]^d\}$. By construction, the sets in $\mathcal{P}_\epsilon$ are disjoint. We first claim the following:

**Lemma 26.** $|\{A \in \mathcal{P}_\epsilon : A \cap \mathcal{M} \neq \emptyset\}| \leq C2^d \epsilon^{-s}$.

*Proof.* Let, $r = \mathcal{N}(\epsilon; \mathcal{M}, \ell_\infty)$ and suppose that $\{\boldsymbol{a}_1, \ldots, \boldsymbol{a}_r\}$ be an $\epsilon$-net of $\mathcal{M}$ and $\mathcal{P}_\epsilon^* = \{B_{\ell_\infty}(\boldsymbol{a}_i, \epsilon) : i \in [r]\}$ be an optimal $\epsilon$-cover of $\mathcal{M}$. Note that each box in $\mathcal{P}_\epsilon^*$ can intersect at most $2^d$ boxes in $\mathcal{P}_\epsilon$. This implies that,

$$|\mathcal{P}_\epsilon \cap \mathcal{M}| \leq |\mathcal{P}_\epsilon \cap (\cup_{i=1}^r B_{\ell_\infty}(\boldsymbol{a}_i, \epsilon))| = |\cup_{i=1}^r (\mathcal{P}_\epsilon \cap B_{\ell_\infty}(\boldsymbol{a}_i, \epsilon))| \leq 2^d r,$$

which concludes the proof. $\qquad\square$

We are now ready to prove Theorem 18. For the ease of readability, we restate the theorem as follows:

**Theorem 18.** *Let $f$ be an element of $\mathcal{H}^\beta(\mathbb{R}^d, \mathbb{R}, C)$, where $C > 0$. Then, for any $s > d_\gamma$, there exists constants $\epsilon_0$ (which may depend on $\gamma$) and $\alpha$, (which may depend on $\beta$, $d$, and $C$), such that if $\epsilon \in (0, \epsilon_0]$, a ReLU network $\hat{f}$ can be constructed with $\mathcal{L}(\hat{f}) \leq \alpha \log(1/\epsilon)$ and $\mathcal{W}(\hat{f}) \leq \alpha \log(1/\epsilon)\epsilon^{-s/\beta}$, satisfying the condition, $\|f - \hat{f}\|_{\mathbb{L}_\infty(\mathcal{M})} \leq \epsilon$.*

*Proof.* We also let $\mathcal{I} = \left\{\boldsymbol{i} \in [K]^d : B_{\ell_\infty}(\boldsymbol{\theta^i}, \epsilon) \cap \mathcal{M} \neq \emptyset\right\}$. We also let $\mathcal{I}^\dagger = \{\boldsymbol{j} \in [K]^d : \min_{\boldsymbol{i} \in \mathcal{I}} \|\boldsymbol{i} - \boldsymbol{j}\|_1 \leq 1\}$. We know that $|\mathcal{I}^\dagger| \leq 3^d |\mathcal{I}| \leq 6^d N(\epsilon; \mathcal{M}, \ell_\infty)$. For $0 < b \leq a$, let,

$$\xi_{a,b}(x) = \text{ReLU}\left(\frac{x+a}{a-b}\right) - \text{ReLU}\left(\frac{x+b}{a-b}\right) - \text{ReLU}\left(\frac{x-b}{a-b}\right) + \text{ReLU}\left(\frac{x-a}{a-b}\right).$$

A pictorial view of this function is given in Fig. 2 and can be implemented by a ReLU network of depth two and width four. Thus, $\mathcal{L}(\xi_{a,b}) = 2$ and $\mathcal{W}(\xi_{a,b}) = 12$. Suppose that $0 < \delta < \epsilon/3$ and let, $\zeta(\boldsymbol{x}) = \prod_{\ell=1}^d \xi_{\epsilon+\delta, \delta}(x_\ell)$. It is easy to observe that $\{\zeta(\cdot - \boldsymbol{\theta^i}) : \boldsymbol{i} \in \mathcal{I}^\dagger\}$ forms a partition of unity on $\mathcal{M}$, i.e. $\sum_{\boldsymbol{i} \in \mathcal{I}^\dagger} \zeta(\boldsymbol{x} - \boldsymbol{\theta^i}) = 1, \forall \boldsymbol{x} \in \mathcal{M}$.

We consider the Taylor approximation of $f$ around $\boldsymbol{\theta}$ as,

$$P_{\boldsymbol{\theta}}(\boldsymbol{x}) = \sum_{|\boldsymbol{s}| < \lfloor \beta \rfloor} \frac{\partial^{\boldsymbol{s}} f(\boldsymbol{\theta})}{\boldsymbol{s}!} (\boldsymbol{x} - \boldsymbol{\theta})^{\boldsymbol{s}}.$$

Note that for any $\boldsymbol{x} \in [0,1]^d$, $f(\boldsymbol{x}) - P_{\boldsymbol{\theta}}(\boldsymbol{x}) = \sum_{\boldsymbol{s}:|\boldsymbol{s}|=\lfloor\beta\rfloor} \frac{(\boldsymbol{x}-\boldsymbol{\theta})^{\boldsymbol{s}}}{\boldsymbol{s}!}(\partial^{\boldsymbol{s}} f(\boldsymbol{y}) - \partial^{\boldsymbol{s}} f(\boldsymbol{\theta}))$, for some $\boldsymbol{y}$, which is a convex combination of $\boldsymbol{x}$ and $\boldsymbol{\theta}$. Thus,

$$
\begin{aligned}
f(\boldsymbol{x}) - P_{\boldsymbol{\theta}}(\boldsymbol{x}) = \sum_{\boldsymbol{s}:|\boldsymbol{s}|=\lfloor\beta\rfloor} \frac{(\boldsymbol{x}-\boldsymbol{\theta})^{\boldsymbol{s}}}{\boldsymbol{s}!}(\partial^{\boldsymbol{s}} f(\boldsymbol{y}) - \partial^{\boldsymbol{s}} f(\boldsymbol{\theta})) \leq& \|\boldsymbol{x}-\boldsymbol{\theta}\|_\infty^{\lfloor\beta\rfloor} \sum_{\boldsymbol{s}:|\boldsymbol{s}|=\lfloor\beta\rfloor} \frac{1}{\boldsymbol{s}!}|\partial^{\boldsymbol{s}} f(\boldsymbol{y}) - \partial^{\boldsymbol{s}} f(\boldsymbol{\theta})| \\
\leq& \|\boldsymbol{x}-\boldsymbol{\theta}\|_\infty^{\lfloor\beta\rfloor}\|\boldsymbol{y}-\boldsymbol{\theta}\|_\infty^{\beta-\lfloor\beta\rfloor} \\
\leq& \|\boldsymbol{x}-\boldsymbol{\theta}\|_\infty^\beta.
\end{aligned} \tag{14}
$$

Next we define $\tilde{f}(\boldsymbol{x}) = \sum_{\boldsymbol{i}\in\mathcal{I}^\dagger} \zeta(\boldsymbol{x}-\boldsymbol{\theta}^{\boldsymbol{i}}) P_{\boldsymbol{\theta}^{\boldsymbol{i}}}(\boldsymbol{x})$. Thus, if $\boldsymbol{x}\in\mathcal{M}$,

$$
\begin{aligned}
|f(\boldsymbol{x}) - \tilde{f}(\boldsymbol{x})| = \left|\sum_{\boldsymbol{i}\in\mathcal{I}^\dagger} \zeta(\boldsymbol{x}-\boldsymbol{\theta}^{\boldsymbol{i}})(f(\boldsymbol{x})-P_{\boldsymbol{\theta}^{\boldsymbol{i}}}(\boldsymbol{x}))\right| \leq& \sum_{\boldsymbol{i}\in\mathcal{I}^\dagger:\|\boldsymbol{x}-\boldsymbol{\theta}^{\boldsymbol{i}}\|_\infty\leq 2\epsilon} |f(\boldsymbol{x})-P_{\boldsymbol{\theta}^{\boldsymbol{i}}}(\boldsymbol{x})| \\
\leq& 2^d (2\epsilon)^\beta \\
=& 2^{d+\beta}\epsilon^\beta.
\end{aligned} \tag{15}
$$

We note that, $\tilde{f}(\boldsymbol{x}) = \sum_{\boldsymbol{i}\in\mathcal{I}^\dagger} \zeta(\boldsymbol{x}-\boldsymbol{\theta}^{\boldsymbol{i}}) P_{\boldsymbol{\theta}^{\boldsymbol{i}}}(\boldsymbol{x}) = \sum_{\boldsymbol{i}\in\mathcal{I}^\dagger}\sum_{|\boldsymbol{s}|<\lfloor\beta\rfloor} \frac{\partial^{\boldsymbol{s}} f(\boldsymbol{\theta}^{\boldsymbol{i}})}{\boldsymbol{s}!}\zeta(\boldsymbol{x}-\boldsymbol{\theta}^{\boldsymbol{i}})\left(\boldsymbol{x}-\boldsymbol{\theta}^{\boldsymbol{i}}\right)^{\boldsymbol{s}}$. Let $a_{\boldsymbol{i},\boldsymbol{s}} = \frac{\partial^{\boldsymbol{s}} f(\boldsymbol{\theta}^{\boldsymbol{i}})}{\boldsymbol{s}!}$ and

$$
\hat{f}_{\boldsymbol{i},\boldsymbol{s}}(\boldsymbol{x}) = \mathrm{prod}_m^{(d+|\boldsymbol{s}|)}(\xi_{\epsilon_1,\delta_1}(x_1-\theta_1^{\boldsymbol{i}}),\ldots,\xi_{\epsilon_d,\delta_d}(x_d-\theta_d^{\boldsymbol{i}}),\underbrace{(x_1-\theta_1^{\boldsymbol{i}}),\ldots,(x_1-\theta_1^{\boldsymbol{i}})}_{s_1 \text{ times}},
$$

$$
\ldots,\underbrace{(x_1-\theta_d^{\boldsymbol{i}}),\ldots,(x_d-\theta_d^{\boldsymbol{i}})}_{s_d \text{ times}}),
$$

where, $\mathrm{prod}(\cdot)$ is defined in Lemma 34. Here, $\mathrm{prod}_m^{(d+|\boldsymbol{s}|)}$ has at most $d+|\boldsymbol{s}| \leq d+\lfloor\beta\rfloor$ many inputs. By Lemma 34, $\mathrm{prod}_m^{(d+|\boldsymbol{s}|)}$ can be implemented by a ReLU network with $\mathcal{L}(\mathrm{prod}_m^{(d+|\boldsymbol{s}|)}),\mathcal{W}(\mathrm{prod}_m^{(d+|\boldsymbol{s}|)}) \leq c_3 m$. Thus, $\mathcal{L}(\hat{f}_{\boldsymbol{i},\boldsymbol{s}}) \leq c_3 m + 2$ and $\mathcal{W}(\hat{f}_{\boldsymbol{i},\boldsymbol{s}}) \leq c_3 m + 8d + 4|\boldsymbol{s}| \leq c_3 m + 8d + 4k$. With this $\hat{f}_{\boldsymbol{i},\boldsymbol{s}}$, we observe that,

$$
\left|\hat{f}_{\boldsymbol{i},\boldsymbol{s}}(\boldsymbol{x}) - \zeta(\boldsymbol{x}-\boldsymbol{\theta}^{\boldsymbol{i}})\left(\boldsymbol{x}-\boldsymbol{\theta}^{\boldsymbol{i}}\right)^{\boldsymbol{s}}\right| \leq \frac{(d+\lfloor\beta\rfloor)^3}{2^{2m+2}}, \forall \boldsymbol{x}\in\mathcal{M}. \tag{16}
$$

Finally, let, $\hat{f}(\boldsymbol{x}) = \sum_{\boldsymbol{i}\in\mathcal{I}^\dagger}\sum_{|\boldsymbol{s}|\leq\lfloor\beta\rfloor} a_{\boldsymbol{i},\boldsymbol{s}}\hat{f}_{\boldsymbol{i},\boldsymbol{s}}(\boldsymbol{x})$. Clearly, $\mathcal{L}(\hat{f}_{\boldsymbol{i},\boldsymbol{s}}) \leq c_3 m + 3$ and $\mathcal{W}(\hat{f}_{\boldsymbol{i},\boldsymbol{s}}) \leq k^d(c_3 m + 8d + 4k)$. This implies that,

$$
\begin{aligned}
|\hat{f}(\boldsymbol{x}) - \tilde{f}(\boldsymbol{x})| \leq& \sum_{\boldsymbol{i}\in\mathcal{I}^\dagger:\|\boldsymbol{x}-\boldsymbol{\theta}^{\boldsymbol{i}}\|_\infty\leq 2\epsilon}\sum_{|\boldsymbol{s}|<\lfloor\beta\rfloor} |a_{\boldsymbol{i},\boldsymbol{s}}||\zeta(\boldsymbol{x}-\boldsymbol{\theta}^{\boldsymbol{i}})\hat{f}_{\boldsymbol{i}\boldsymbol{s}}(\boldsymbol{x}) - \left(\boldsymbol{x}-\boldsymbol{\theta}^{\boldsymbol{i}}\right)^{\boldsymbol{s}}| \\
\leq& 2^d \sum_{|\boldsymbol{s}|<\lfloor\beta\rfloor} |a_{\boldsymbol{\theta},\boldsymbol{s}}|\left|\hat{f}_{\boldsymbol{\theta}^{i(\boldsymbol{x})},\boldsymbol{s}}(\boldsymbol{x}) - \zeta_{\epsilon,\delta}(\boldsymbol{x}-\boldsymbol{\theta}^{(i(\boldsymbol{x}))})\left(\boldsymbol{x}-\boldsymbol{\theta}^{i(\boldsymbol{x})}\right)^{\boldsymbol{s}}\right| \\
\leq& \frac{(d+\lfloor\beta\rfloor)^3 C}{2^{2m+2-d}}.
\end{aligned}
$$

We thus get that if $\boldsymbol{x}\in\mathcal{M}$,

$$
|f(\boldsymbol{x}) - \hat{f}(\boldsymbol{x})| \leq |f(\boldsymbol{x})-\tilde{f}(\boldsymbol{x})| + |\hat{f}(\boldsymbol{x})-\tilde{f}(\boldsymbol{x})| \leq 2^{d+\beta}\epsilon^\beta + \frac{(d+\lfloor\beta\rfloor)^3 C}{2^{2m+2-d}}. \tag{17}
$$

We choose $\epsilon = \left(\frac{\eta}{2^{d+k+2}}\right)^{1/\beta}$ and $m = \left\lceil\log_2\left(\frac{(d+k)^3 C}{\eta}\right)\right\rceil + d - 1$. Then,

$$
\|f-\hat{f}\|_{\mathbb{L}_\infty(\mathcal{M})} \leq \eta.
$$

We note that $\hat{f}$ has $|\mathcal{I}^\dagger| \leq 6^d N_\epsilon(\mathcal{M}) \lesssim 6^d\epsilon^{-s}$ many networks with depth $c_3 m + 3$ and number of weights $\lfloor\beta\rfloor^d(c_3 m + 8d + 4\lfloor\beta\rfloor)$. Thus, $\mathcal{L}(\hat{f}) \leq c_3 m + 4$ and $\mathcal{W}(\hat{f}) \leq \epsilon^{-s}(6\lfloor\beta\rfloor)^d(c_3 m + 8d + 4\lfloor\beta\rfloor)$. we thus get,

$$
\mathcal{L}(\hat{f}) \leq c_3 m + 4 \leq c_3\left(\left\lceil\log_2\left(\frac{(d+\lfloor\beta\rfloor)^3 C_\delta}{\eta}\right)\right\rceil + d - 1\right) + 4 \leq c_4\log\left(\frac{1}{\eta}\right),
$$

where $c_4$ is a function of $\delta$, $\lfloor \beta \rfloor$ and $d$. Similarly,

$$
\begin{aligned}
\mathcal{W}(\hat{f}) \leq &\epsilon^{-s}(6\lfloor\beta\rfloor)^d (c_3 m + 8d + 4\lfloor\beta\rfloor) \\
\leq &\left(\frac{\eta}{2^{d+k+2}}\right)^{-s/\beta} (6\lfloor\beta\rfloor)^d \left(c_3 \left(\log_2\left(\frac{(d+\lfloor\beta\rfloor)^3 C_\delta}{\eta}\right) + d - 1\right) + 8d + 4\lfloor\beta\rfloor\right) \\
\leq &c_6 \log(1/\eta)\eta^{-s/\beta}.
\end{aligned}
$$

Taking $\alpha = c_4 \vee c_6$ gives the result. $\qquad\square$

### C.3.2 Proof of Lemma 19

**Lemma 19.** *Suppose assumptions A1–3 hold and let, $\mathrm{diss}(\cdot,\cdot) \equiv \mathcal{W}_1(\cdot,\cdot)$ or $MMD_{\mathcal{K}}^2(\cdot,\cdot)$. Also, let $s > d_\mu$. Then, we can find positive constants $\epsilon_0$, $\alpha_0$ and $R$, that might depend on $d, \ell, \tilde{G}$ and $\tilde{E}$, such that if $0 < \epsilon_g, \epsilon_e \leq \epsilon_0$ and $\mathcal{G} = \mathcal{RN}(W_g, L_g, R)$ and $\mathcal{E} = \mathcal{RN}(W_e, L_e, R)$, with*

$$L_e \leq \alpha_0 \log(1/\epsilon_g),\ L_g \leq \alpha_0 \log(1/\alpha_g),\ W_e \leq \alpha_0 \epsilon_e^{-s/\alpha_e} \log(1/\epsilon_e)\ \text{and}\ W_g \leq \alpha_0 \epsilon_g^{-\ell/\alpha_g} \log(1/\epsilon_g)$$

*then, $\Delta_{miss} \lesssim \epsilon_g + \epsilon_e^{\alpha_g \wedge 1}$.*

*Proof.* We first prove the result for the Wasserstein-1 distance and then for the $\mathrm{MMD}_{\mathcal{K}}$-metric.

**Case 1:** $\mathrm{diss}(\cdot,\cdot) \equiv \mathcal{W}_1(\cdot,\cdot)$    For any $G \in \mathcal{G}$ and $E \in \mathcal{E}$, we observe that,

$$
\begin{aligned}
V(\mu,\nu,G,E) \leq &V(\mu,\nu,\tilde{G},\tilde{E}) + |V(\mu,\nu,G,E) - V(\mu,\nu,\tilde{G},\tilde{E})| \\
\leq &\|c(\cdot,\tilde{G}\circ\tilde{E}(\cdot)) - c(\cdot,G\circ E(\cdot))\|_{\mathbb{L}_\infty(\mathcal{M})} + |\mathcal{W}_1(E_\sharp\mu,\nu) - \mathcal{W}_1(\tilde{E}_\sharp\mu,\nu)| \\
\lesssim &\|G\circ E - \tilde{G}\circ\tilde{E}\|_{\mathbb{L}_\infty(\mathcal{M})} + \mathcal{W}_1(\tilde{E}_\sharp\mu, \mathbb{E}_\sharp\mu) \\
\lesssim &\|G\circ E - \tilde{G}\circ\tilde{E}\|_{\mathbb{L}_\infty(\mathrm{supp}(\mu))} + \|\tilde{E} - \mathbb{E}\|_{\mathbb{L}_\infty(\mathrm{supp}(\mu))} \\
\leq &\|G\circ E - \tilde{G}\circ E\|_{\mathbb{L}_\infty(\mathrm{supp}(\mu))} + \|\tilde{G}\circ E - \tilde{G}\circ\tilde{E}\|_{\mathbb{L}_\infty(\mathrm{supp}(\mu))} + \|\tilde{E} - \mathbb{E}\|_{\mathbb{L}_\infty(\mathrm{supp}(\mu))} \\
\lesssim &\|G - \tilde{G}\|_{\mathbb{L}_\infty([0,1]^\ell)} + \|E - \tilde{E}\|_{\mathbb{L}_\infty(\mathrm{supp}(\mu))}^{\alpha_g\wedge 1} + \|\tilde{E} - \mathbb{E}\|_{\mathbb{L}_\infty(\mathrm{supp}(\mu))}
\end{aligned}
$$

We can take $\mathcal{G} = \mathcal{RN}(\log(1/\epsilon_g), \epsilon_g^{-\ell/\alpha_g}\log(1/\epsilon_g))$ and $\mathcal{E} = \mathcal{RN}(\log(1/\epsilon_e), \epsilon_e^{-s/\alpha_e}\log(1/\epsilon_e))$ by approximating in each of the individual coordinate-wise output of the vector-valued functions $\tilde{G}$ and $\tilde{E}$ and stacking them parallelly. This makes,

$$V(\mu,\nu,G,E) \lesssim \epsilon_g + \epsilon_e^{\alpha_g\wedge 1} + \epsilon_e \lesssim \epsilon_g + \epsilon_e^{\alpha_g\wedge 1}.$$

**Case 2:** $\mathrm{diss}(\cdot,\cdot) \equiv \mathrm{MMD}_k^2(\cdot,\cdot)$    Before we begin, we note that,

$$
\begin{aligned}
&|\mathrm{MMD}_{\mathcal{K}}^2(E_\sharp\mu,\nu) - \mathrm{MMD}_{\mathcal{K}}^2(\tilde{E}_\sharp\mu,\nu)| \\
\leq &|\mathbb{E}_{X\sim\mu,X'\sim\mu}\mathcal{K}(E(X),E(X')) - \mathbb{E}_{X\sim\mu,X'\sim\mu}\mathcal{K}(\tilde{E}(X),\tilde{E}(X'))| \\
&+ 2|\mathbb{E}_{X\sim\mu,Z\sim\nu}\mathcal{K}(E(X),Z) - \mathbb{E}_{X\sim\mu,Z\sim\nu}\mathcal{K}(\tilde{E}(X),Z)| \\
\leq &2\tau_k\|E - \tilde{E}\|_{\mathbb{L}_\infty(\mathrm{supp}(\mu))} + 2\tau_k\|E - \tilde{E}\|_{\mathbb{L}_\infty(\mathrm{supp}(\mu))} \\
= &4\tau_k\|E - \tilde{E}\|_{\mathbb{L}_\infty(\mathrm{supp}(\mu))}. \qquad\qquad\qquad (18)
\end{aligned}
$$

For any $G \in \mathcal{G}$ and $E \in \mathcal{E}$, we observe that,

$$
\begin{aligned}
V(\mu,\nu,G,E) = &V(\mu,\nu,\tilde{G},\tilde{E}) + |V(\mu,\nu,G,E) - V(\mu,\nu,\tilde{G},\tilde{E})| \\
= &\|c(\cdot,\tilde{G}\circ\tilde{E}(\cdot)) - c(\cdot,G\circ E(\cdot))\|_{\mathbb{L}_\infty(\mathrm{supp}(\mu))} + |\mathrm{MMD}_{\mathcal{K}}^2(E_\sharp\mu,\nu) - \mathrm{MMD}_{\mathcal{K}}^2(\tilde{E}_\sharp\mu,\nu)| \\
\lesssim &\|G\circ E - \tilde{G}\circ\tilde{E}\|_{\mathbb{L}_\infty(\mu)} + 4\tau_k\|E - \tilde{E}\|_{\mathbb{L}_\infty(\mathrm{supp}(\mu))} \qquad\qquad (19) \\
\lesssim &\|G\circ E - \tilde{G}\circ\tilde{E}\|_{\mathbb{L}_\infty(\mathrm{supp}(\mu))} + \|\tilde{E} - \mathbb{E}\|_{\mathbb{L}_\infty(\mathrm{supp}(\mu))} \\
\leq &\|G\circ E - \tilde{G}\circ E\|_{\mathbb{L}_\infty(\mathrm{supp}(\mu))} + \|\tilde{G}\circ E - \tilde{G}\circ\tilde{E}\|_{\mathbb{L}_\infty(\mathrm{supp}(\mu))} + \|\tilde{E} - \mathbb{E}\|_{\mathbb{L}_\infty(\mathrm{supp}(\mu))} \\
\lesssim &\|G - \tilde{G}\|_{\mathbb{L}_\infty([0,1]^\ell)} + \|E - \tilde{E}\|_{\mathbb{L}_\infty(\mathrm{supp}(\mu))}^{\alpha_g\wedge 1} + \|\tilde{E} - \mathbb{E}\|_{\mathbb{L}_\infty(\mathrm{supp}(\mu))}.
\end{aligned}
$$

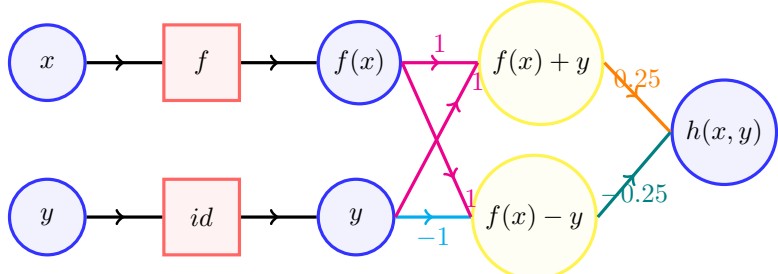

Figure 3: A representation of the network $h(\cdot,\cdot)$. The magenta lines represent $d'$ weights of value 1. Similarly, cyan lines represent $d'$ weights of value $-1$. Finally, the orange and teal lines represent $d'$ weights (each) with values $+0.25$ and $-0.25$, respectively. The identity map takes $2d'\mathcal{L}(f)$ many weights (see remark 15 (iv) of Nakada & Imaizumi (2020)). The magenta, cyan, orange and teal connections take $6d'$ many weights. All activations are taken to be ReLU, except the output of the yellow nodes, whose activation is $\sigma(x) = x^2$.

In the above calculations, we have used (18) to arrive at (19). As before, we take $\mathcal{G} = \mathcal{RN}(\log(1/\epsilon_g), \epsilon_g^{-\ell/\alpha_g}\log(1/\epsilon_g))$ and $\mathcal{E} = \mathcal{RN}(\log(1/\epsilon_e), \epsilon_e^{-s/\alpha_e}\log(1/\epsilon_e))$ by approximating in each of the individual coordinate-wise output of the vector-valued functions $\tilde{G}$ and $\tilde{E}$ and stacking them parallelly. This makes,

$$V(\mu,\nu,G,E) \lesssim \epsilon_g + \epsilon_e^{\alpha_g \wedge 1} + \epsilon_e \lesssim \epsilon_g + \epsilon_e^{\alpha_g \wedge 1}.$$

$\square$

## C.4 Proofs from Section B.2

### C.4.1 Proof of Lemma 20

**Lemma 20.** *Suppose that $n \geq 6$ and $\mathcal{F}$ are a class neural network with depth at most $L$ and number of weights at most $W$. Furthermore, the activation functions are piece-wise polynomial activation with the number of pieces and degree at most $k \in \mathbb{N}$. Then, there is a constant $\theta$ (that might depend on $d$ and $d'$), such that, if $n \geq \theta(W + 6d' + 2d'L)(L+3)(\log(W + 6d' + 2d'L) + L + 3)$,*

$$\log\mathcal{N}(\epsilon; \mathcal{F}_{|_{X_{1:n}}}, \ell_\infty) \lesssim (W + 6d' + 2d'L)(L+3)(\log(W + 6d' + 2d'L) + L + 3)\log\left(\frac{nd'}{\epsilon}\right),$$

*where $d'$ is the output dimension of the networks in $\mathcal{F}$.*

*Proof.* We let, $h(x,y) = y^\top f(x)$ and let $\mathcal{H} = \{h(x,y) = y^\top f(x) : f \in \mathcal{F}\}$. Also, let, $\mathcal{T} = \{(h(X_i, e_\ell)|_{i \in [n], \ell \in [d']}) \in \mathbb{R}^{nd'} : h \in \mathcal{H}\}$. Here $e_\ell$ denotes the $\ell$-th unit vector. By construction of $\mathcal{T}$, it is clear that, $\mathcal{N}(\epsilon; \mathcal{F}_{|_{X_{1:n}}}, \ell_\infty) = \mathcal{N}(\epsilon; \mathcal{T}, \ell_\infty)$. We observe that,

$$h(x,y) = \frac{1}{4}(\|y + f(x)\|_2^2 - \|y - f(x)\|_2^2)$$

Clearly, $h$ can be implemented by a network with $\mathcal{L}(h) = \mathcal{L}(f) + 3$ and $\mathcal{W}(h) = \mathcal{W}(f) + 6d' + 2d'\mathcal{L}(f)$ (see Fig. 3 for such a construction). Thus, from Theorem 12.9 of Anthony & Bartlett (2009) (see Lemma 37), we note that, if $n \geq \mathrm{Pdim}(\mathcal{H})$,

$$\mathcal{N}(\epsilon; \mathcal{T}, \ell_\infty) \leq \left(\frac{2end'}{\epsilon\mathrm{Pdim}(\mathcal{H})}\right)^{\mathrm{Pdim}(\mathcal{H})},$$

with,

$$\mathrm{Pdim}(\mathcal{H}) \lesssim \mathcal{W}(h)\mathcal{L}(h)\log\mathcal{W}(h) + \mathcal{W}(h)\mathcal{L}^2(h),$$

from applying Theorem 6 of Bartlett et al. (2019) (see Lemma 38). This implies that,

$$
\begin{aligned}
\log \mathcal{N}(\epsilon; \mathcal{H}, \ell_\infty) &\leq \mathrm{Pdim}(\mathcal{H}) \log \left( \frac{2end'}{\epsilon \mathrm{Pdim}(\mathcal{H})} \right) \\
&\leq \mathrm{Pdim}(\mathcal{H}) \log \left( \frac{nd'}{\epsilon} \right) \\
&\lesssim \left( \mathcal{W}(h)\mathcal{L}(h) \log \mathcal{W}(h) + \mathcal{W}(h)\mathcal{L}^2(h) \right) \log \left( \frac{nd'}{\epsilon} \right).
\end{aligned}
$$

Plugging in the values of $\mathcal{W}(h)$ and $\mathcal{L}(h)$ yields the result. $\qquad \square$

### C.4.2 Proof of Corollary 21

**Corollary 21.** *Suppose that* $\mathcal{W}(\mathcal{E}) \leq W_e$, $\mathcal{L}(\mathcal{E}) \leq L_e$, $\mathcal{W}(\mathcal{G}) \leq W_g$ *and* $\mathcal{L}(\mathcal{G}) \leq L_g$, *with* $L_e, L_g \geq 3$, $W_e \geq 6\ell + 2\ell L_e$ *and* $W_g \geq 6d + 2dL_g$. *Then, there is a constant* $\xi_1$, *such that if* $n \geq \xi_1(W_e + W_g)(L_e + L_g)\left(\log(W_e + W_g) + L_e + L_g\right)$,

$$
\log \mathcal{N}\left(\epsilon; \mathcal{E}_{|X_{1:n}}, \ell_\infty\right) \lesssim W_e L_e (\log W_e + L_e) \log \left( \frac{n\ell}{\epsilon} \right),
$$

$$
\log \mathcal{N}\left(\epsilon; (\mathcal{G} \circ \mathcal{E})_{|X_{1:n}}, \ell_\infty\right) \lesssim (W_e + W_g)(L_e + L_g)\left(\log(W_e + W_g) + L_e + L_g\right) \log \left( \frac{nd}{\epsilon} \right).
$$

*Proof.* The proof easily follows from applying Lemma 20 and noting the sizes of the networks in $\mathcal{E}$ and $\mathcal{G} \circ \mathcal{E}$. $\qquad \square$

### C.4.3 Proof of Lemma 22

**Lemma 22.** *Suppose* $\mathcal{R}(\mathcal{G}) \lesssim 1$ *and* $\mathcal{F} = \{f(x) = c(x, G \circ E(x)) : G \in \mathcal{G}, E \in \mathcal{E}\}$. *Furthermore, let,* $\mathcal{L}(\mathcal{E}) \leq L_e$, $\mathcal{W}(\mathcal{G}) \leq W_g$ *and* $\mathcal{L}(\mathcal{G}) \leq L_g$, *with* $L_e, L_g \geq 3$, $W_e \geq 6\ell + 2\ell L_e$ *and* $W_g \geq 6d + 2dL_g$. *Then, there is a constant* $\xi_2$, *such that if* $n \geq \xi_2(W_e + W_g)(L_e + L_g)\left(\log(W_e + W_g) + L_e + L_g\right)$

$$
\mathbb{E}\|\hat{\mu}_n - \mu\|_{\mathcal{F}} \lesssim n^{-1/2}\left((W_e + W_g)(L_e + L_g)\left(\log(W_e + W_g) + L_e + L_g\right) \log(nd)\right)^{1/2}.
$$

*Proof.* Let $\mathcal{R}(\mathcal{G}) \leq B$, for some $B > 0$ and let $B_c = \sup_{0 \leq x \leq B} |c(x)|$ From Dudley's chaining (Wainwright, 2019, Theorem 5.22),

$$
\mathbb{E}\|\hat{\mu}_n - \mu\|_{\mathcal{F}} \lesssim \mathbb{E}_{X_{1:n}} \inf_{0 \leq \delta \leq B_c/2} \left( \delta + \frac{1}{\sqrt{n}} \int_\delta^{B_c/2} \sqrt{\log \mathcal{N}(\epsilon, \mathcal{F}_{|X_{1:n}}, \ell_\infty)} d\epsilon \right). \tag{20}
$$

Let For any $G \in \mathcal{G}$ and $E \in \mathcal{E}$, we can find $v \in \mathcal{C}(\epsilon; (\mathcal{G} \circ \mathcal{E})_{|X_{1:n}}, \ell_\infty)$, such that, $\|(G \circ E)_{|X_{1:n}} - v\|_\infty \leq \epsilon$. This implies that $\|(G \circ E)(X_i) - v_i\|_\infty \leq \epsilon$, for all $i \in [n]$. Let, $\mathcal{A} = \{(c(X_1, v_1), \ldots, c(X_n, v_n)) : v \in (\mathcal{G} \circ \mathcal{E})_{|X_{1:n}}\}$. Thus, For any $G \in \mathcal{G}$, $E \in \mathcal{E}$,

$$
\max_{1 \leq i \leq n} |c(X_i, G \circ E(X_i)) - c(X_i, v_i)| \leq \tau_c \max_{1 \leq i \leq n} \|(G \circ E)(X_i) - v_i\|_\infty \leq \tau_c \epsilon.
$$

Thus, $\mathcal{A}$ constitutes a $\tau_c\epsilon$-cover of $\mathcal{F}_{|X_{1:n}}$. Hence,

$$
\mathcal{N}(\epsilon, \mathcal{F}_{|X_{1:n}}, \ell_\infty) \leq \mathcal{N}(\epsilon/\tau_c, (\mathcal{G} \circ \mathcal{E})_{|X_{1:n}}, \ell_\infty) \leq (W_e + W_g)(L_e + L_g)\left(\log(W_e + W_g) + L_e + L_g\right) \log \left( \frac{\tau_c nd}{\epsilon} \right).
$$

Here, the last inequality follows from Lemma 20. Plugging in the above bound in equation (20), we get,

$$
\mathbb{E}\|\hat{\mu}_n - \mu\|_{\mathcal{F}} \lesssim \mathbb{E}_{X_{1:n}} \inf_{0 \leq \delta \leq B_c/2} \left( \delta + \frac{1}{\sqrt{n}} \int_\delta^{B_c/2} \sqrt{\log \mathcal{N}(\epsilon, \mathcal{F}_{|X_{1:n}}, \ell_\infty)} d\epsilon \right)
$$

$$\leq \sqrt{\frac{(W_e + W_g)(L_e + L_g)\log(W_e + W_g)}{n}} \int_0^{B_c} \sqrt{\log\left(\frac{\tau_c n d}{\epsilon}\right)} d\epsilon$$

$$\lesssim \sqrt{\frac{(W_e + W_g)(L_e + L_g)\left(\log(W_e + W_g) + L_e + L_g\right)\log(nd)}{n}}.$$

$\square$

### C.4.4  PROOF OF LEMMA 23

**Lemma 23.** *Let $\hat{\mu}_n = \frac{1}{n}\sum_{i=1}^n \delta_{X_i}$ and $\mathcal{E} = \mathcal{RN}(L_e, W_e)$. Then,*

$$\sup_{E \in \mathcal{E}} |\mathcal{W}_1(E_\sharp \mu_n, \nu) - \mathcal{W}_1(E_\sharp \mu, \nu)| \lesssim \left(n^{-1/\ell} \vee n^{-1/2}\log n\right) + \sqrt{\frac{W_e L_e(\log W_e + L_e)\log(n\ell)}{n}}.$$

*Furthermore,*

$$\sup_{E \in \mathcal{E}} |\mathcal{W}_1(E_\sharp \hat{\mu}_n, \hat{\nu}_m) - \mathcal{W}_1(E_\sharp \mu, \nu)| \lesssim \left(n^{-1/\ell} \vee n^{-1/2}\log n\right) + \left(m^{-1/\ell} \vee m^{-1/2}\log m\right)$$

$$+ \sqrt{\frac{W_e L_e(\log W_e + L_e)\log(n\ell)}{n}}.$$

*Proof.* Note that if $\mathrm{diss}(\cdot,\cdot) = \mathcal{W}_1(\cdot,\cdot)$, then

$$\sup_{E \in \mathcal{E}} |\widehat{\mathrm{diss}}(E_\sharp \mu, \nu) - \mathrm{diss}(E_\sharp \mu, \nu)| = \sup_{E \in \mathcal{E}} |\mathcal{W}_1(E_\sharp \hat{\mu}_n, \nu) - \mathcal{W}_1(E_\sharp \mu, \nu)| \leq \sup_{E \in \mathcal{E}} \mathcal{W}_1(E_\sharp \hat{\mu}_n, E_\sharp \mu)$$

We note that,

$$\sup_{E \in \mathcal{E}} W(E_\sharp \hat{\mu}, E_\sharp \mu) = \sup_{E \in \mathcal{E}} \sup_{f : \|f\|_{\mathrm{Lip}} \leq 1} \mathbb{E}_{X \sim \mu, \hat{X} \sim \hat{\mu}} f(E(X)) - f(E(\hat{X}))$$

We take $\mathcal{F}_1 = \{f : [0,1]^\ell \to \mathbb{R} : \|f\|_{\mathrm{Lip}} \leq 1\} = \mathcal{H}^1(\sqrt{\ell})$. By the result of Kolmogorov & Tikhomirov (1961) (Lemma 36), we note that $\log \mathcal{N}(\epsilon; \mathcal{F}_1, \ell_\infty) \lesssim \epsilon^{-\ell}$. Furthermore, if we take $\mathcal{F}_2 = \mathcal{E}$, we observe that, $\log \mathcal{N}(\epsilon; \mathcal{E}_{|X_{1:n}}, \ell_\infty) \lesssim W_e L_e(\log W_e + L_e)\log\left(\frac{n\ell}{\epsilon}\right)$ from Lemma 21. From Dudley's chaining, we observe the following:

$$\mathbb{E}\sup_{E \in \mathcal{E}} W(E_\sharp \hat{\mu}, E_\sharp \mu)$$

$$= \mathbb{E}\sup_{E \in \mathcal{E}} \sup_{f : \|f\|_{\mathrm{Lip}} \leq 1} \mathbb{E}_{X \sim \mu, \hat{X} \sim \hat{\mu}} f(E(X)) - f(E(\hat{X}))$$

$$= \mathbb{E}\|\hat{\mu} - \mu\|_{\mathcal{F}_1 \circ \mathcal{E}}$$

$$\lesssim \mathbb{E}\inf_{0 \leq \delta \leq R_e} \left(\delta + \frac{1}{\sqrt{n}} \int_\delta^{R_e} \sqrt{\log \mathcal{N}\left(\epsilon; (\mathcal{F}_1 \circ \mathcal{E})_{|X_{1:n}}, \ell_\infty\right)} d\epsilon\right)$$

$$\leq \mathbb{E}\inf_{0 \leq \delta \leq R_e} \left(\delta + \frac{1}{\sqrt{n}} \int_\delta^{R_e} \sqrt{\log \mathcal{N}(\epsilon/2; \mathcal{F}_1, \ell_\infty) + \log \mathcal{N}\left(\epsilon/2; \mathcal{E}_{|X_{1:n}}, \ell_\infty\right)} d\epsilon\right)$$

$$\lesssim \mathbb{E}\inf_{0 \leq \delta \leq R_e} \left(\delta + \frac{1}{\sqrt{n}} \int_\delta^{R_e} \left(\sqrt{\log \mathcal{N}(\epsilon/2; \mathcal{F}_1, \ell_\infty)} + \sqrt{\log \mathcal{N}\left(\epsilon/2; \mathcal{E}_{|X_{1:n}}, \ell_\infty\right)}\right) d\epsilon\right)$$

$$\lesssim \mathbb{E}\inf_{0 \leq \delta \leq R_e} \left(\delta + \frac{1}{\sqrt{n}} \int_\delta^{R_e} \sqrt{\log \mathcal{N}(\epsilon/2; \mathcal{F}_1, \ell_\infty)} d\epsilon + \frac{1}{\sqrt{n}} \int_\delta^{R_e} \sqrt{\log \mathcal{N}\left(\epsilon/2; \mathcal{E}_{|X_{1:n}}, \ell_\infty\right)} d\epsilon\right)$$

$$\lesssim \inf_{0 \leq \delta \leq R_e} \left(\delta + \frac{1}{\sqrt{n}} \int_\delta^{R_e} \epsilon^{-\ell/2} d\epsilon + \frac{1}{\sqrt{n}} \int_0^1 \sqrt{W_e L_e(\log W_e + L_e)\log\left(\frac{2en\ell}{\epsilon}\right)} d\epsilon\right)$$

$$\lesssim \inf_{0 \leq \delta \leq R_e} \left(\delta + \frac{1}{\sqrt{n}} \int_\delta^{R_e} \epsilon^{-\ell/2} d\epsilon\right) + \sqrt{\frac{W_e L_e(\log W_e + L_e)\log(n\ell)}{n}}$$

$$\lesssim \inf_{0 \le \delta \le R_e} \left( \delta + \frac{1}{\sqrt{n}} \int_{\delta}^{R_e} \epsilon^{-\ell/2} d\epsilon \right) + \sqrt{\frac{\ell W_e L_e \log W_e \log n}{n}}$$

$$\lesssim \left( n^{-1/\ell} \vee n^{-1/2} \log n \right) + \sqrt{\frac{W_e L_e (\log W_e + L_e) \log(n\ell)}{n}}.$$

$\square$

### C.4.5 PROOF OF LEMMA 24

**Lemma 24.** *Suppose assumption A2 holds and let, $\mathcal{L}(\mathcal{E}) \le L_e$ and $\mathcal{L}(\mathcal{G}) \le L_g$, with $L_e \ge 3$, $W_e \ge 2\ell(3 + L_e)$. Also suppose that, $\Phi = \{\phi \in \mathbb{H}_{\mathcal{K}} : \|\phi\|_{\mathbb{H}_{\mathcal{K}}} \le 1\}$, then,*

$$\mathcal{R}((\Phi \circ \mathcal{E}), X_{1:n}) \lesssim \sqrt{\frac{W_e L_e (\log W_e + L_e) \log(n\ell)}{n}}.$$

*Proof.*

$$\mathcal{R}((\Phi \circ \mathcal{E}), X_{1:n})$$

$$= \frac{1}{n} \mathbb{E} \sup_{\phi \in \Phi, f \in \mathcal{E}} \left| \sum_{i=1}^{n} \sigma_i \phi(f(X_i)) \right|$$

$$= \frac{1}{n} \mathbb{E} \sup_{\phi \in \Phi, f \in \mathcal{E}} \left| \sum_{i=1}^{n} \sigma_i \langle \mathcal{K}(f(X_i), \cdot), \phi \rangle \right|$$

$$= \frac{1}{n} \mathbb{E} \sup_{\phi \in \Phi, f \in \mathcal{E}} \left| \left\langle \sum_{i=1}^{n} \sigma_i \mathcal{K}(f(X_i), \cdot), \phi \right\rangle \right|$$

$$\le \frac{1}{n} \mathbb{E} \sup_{f \in \mathcal{E}} \left\| \sum_{i=1}^{n} \sigma_i \mathcal{K}(f(X_i), \cdot) \right\|_{\mathbb{H}_{\mathcal{K}}}$$

$$= \frac{1}{n} \mathbb{E} \sup_{\boldsymbol{v} \in \mathcal{C}\left(\epsilon, \mathcal{E}_{|X_{1:n}}, \ell_\infty\right)} \sup_{f \in \mathcal{E}} \left\| \sum_{i=1}^{n} \sigma_i \left( \mathcal{K}(v_i, \cdot) + \mathcal{K}(f(X_i), \cdot) - \mathcal{K}(v_i, \cdot) \right) \right\|_{\mathbb{H}_{\mathcal{K}}}$$

$$\le \frac{1}{n} \mathbb{E} \sup_{\boldsymbol{v} \in \mathcal{C}\left(\epsilon, \mathcal{E}_{|X_{1:n}}, \ell_\infty\right)} \sup_{f \in \mathcal{E}} \left( \left\| \sum_{i=1}^{n} \sigma_i \mathcal{K}(v_i, \cdot) \right\|_{\mathbb{H}_{\mathcal{K}}} + \frac{1}{n} \sum_{i=1}^{n} \| \mathcal{K}(f(X_i), \cdot) - \mathcal{K}(v_i, \cdot) \|_{\mathbb{H}_{\mathcal{K}}} \right)$$

$$\le \frac{1}{n} \mathbb{E} \max_{\boldsymbol{v} \in \mathcal{C}\left(\epsilon, \mathcal{E}_{|X_{1:n}}, \ell_\infty\right)} \left\| \sum_{i=1}^{n} \sigma_i \mathcal{K}(v_i, \cdot) \right\|_{\mathbb{H}_{\mathcal{K}}} + \sqrt{2\tau_k \epsilon} \qquad (21)$$

For any $\boldsymbol{v} \in \mathcal{C}\left(\epsilon, \mathcal{E}_{|X_{1:n}}, \ell_\infty\right)$, let, $Y_{\boldsymbol{v}} = \|\sum_{i=1}^{n} \sigma_i \mathcal{K}(v_i, \cdot)\|_{\mathbb{H}_{\mathcal{K}}}$ and $K_{\boldsymbol{v}} = ((\mathcal{K}(v_i, v_j)) \in \mathbb{R}^{n \times n}$. It is easy to observe that, $Y_{\boldsymbol{v}}^2 = \boldsymbol{\sigma}^\top K_{\boldsymbol{v}} \boldsymbol{\sigma}$ and $Y_{\boldsymbol{v}} = \|K_{\boldsymbol{v}}^{1/2} \boldsymbol{\sigma}\|$. By Theorem 2.1 of Rudelson & Vershynin (2013), we note that,

$$\mathbb{P}\left( \left| \|K_{\boldsymbol{v}}^{1/2} \boldsymbol{\sigma}\| - \|K_{\boldsymbol{v}}^{1/2}\|_{\text{HS}} \right| > t \right) \le 2 \exp\left\{ -\frac{ct^2}{\|K_{\boldsymbol{v}}^{1/2}\|^2} \right\} = 2 \exp\left\{ -\frac{ct^2}{\|K_{\boldsymbol{v}}\|} \right\},$$

for some universal constant $c > 0$. From Perron–Frobenius theorem, we note that,

$$\|K_{\boldsymbol{v}}\| \le \max_{1 \le i \le n} \sum_{j=1}^{n} \mathcal{K}(v_i, v_j) \le B^2 n.$$

Hence,

$$\mathbb{P}\left( \left| \|K_{\boldsymbol{v}}^{1/2} \boldsymbol{\sigma}\| - \|K_{\boldsymbol{v}}^{1/2}\|_{\text{HS}} \right| > t \right) \le 2 \exp\left\{ -\frac{ct^2}{nB^2} \right\}.$$

This implies that,

$$\exp(\lambda(\|K_{\boldsymbol{v}}^{1/2}\boldsymbol{\sigma}\| - \|K_{\boldsymbol{v}}^{1/2}\|_{\mathrm{HS}})) \leq \exp\left\{-\frac{c'\lambda^2}{n}\right\},$$

for some absolute constant $c'$, by applying Proposition 2.5.2 of Vershynin (2018). From Theorem 2.5 of Boucheron et al. (2013), we observe that,

$$\mathbb{E}\max_{\boldsymbol{v}\in\mathcal{C}\left(\epsilon,\mathcal{E}_{|X_{1:n}},\ell_\infty\right)}(\|K_{\boldsymbol{v}}^{1/2}\boldsymbol{\sigma}\| - \|K_{\boldsymbol{v}}^{1/2}\|_{\mathrm{HS}}) \lesssim \sqrt{n\log\mathcal{N}\left(\epsilon,\mathcal{E}_{|X_{1:n}},\ell_\infty\right)}.$$

From equation (21), we observe that,

$$
\begin{aligned}
\mathcal{R}(\Phi\circ\mathcal{E}, X_{1:n}) \leq& \frac{1}{n}\mathbb{E}\max_{\boldsymbol{v}\in\mathcal{C}\left(\epsilon,\mathcal{E}_{|X_{1:n}},\ell_\infty\right)}\left\|\sum_{i=1}^n\sigma_i\mathcal{K}(v_i,\cdot)\right\|_{\mathbb{H}_\mathcal{K}} + \sqrt{2\tau_k\epsilon} \\
=& \frac{1}{n}\mathbb{E}\max_{\boldsymbol{v}\in\mathcal{C}\left(\epsilon,\mathcal{E}_{|X_{1:n}},\ell_\infty\right)}(\|K_{\boldsymbol{v}}^{1/2}\boldsymbol{\sigma}\| - \|K_{\boldsymbol{v}}^{1/2}\|_{\mathrm{HS}} + \|K_{\boldsymbol{v}}^{1/2}\|_{\mathrm{HS}}) + \sqrt{2\tau_k\epsilon} \\
\leq& \frac{1}{n}\mathbb{E}\max_{\boldsymbol{v}\in\mathcal{C}\left(\epsilon,\mathcal{E}_{|X_{1:n}},\ell_\infty\right)}(\|K_{\boldsymbol{v}}^{1/2}\boldsymbol{\sigma}\| - \|K_{\boldsymbol{v}}^{1/2}\|_{\mathrm{HS}}) \\
&+ \frac{1}{n}\max_{\boldsymbol{v}\in\mathcal{C}\left(\epsilon,\mathcal{E}_{|X_{1:n}},\ell_\infty\right)}\|K_{\boldsymbol{v}}^{1/2}\|_{\mathrm{HS}} + \sqrt{2\tau_k\epsilon} \\
\lesssim& \sqrt{\frac{\log\mathcal{N}\left(\epsilon,\mathcal{E}_{|X_{1:n}},\ell_\infty\right)}{n}} + \frac{B}{\sqrt{n}} + \sqrt{\epsilon} \\
\lesssim& \sqrt{\frac{W_eL_e\log W_e\log\left(\frac{n\ell}{\epsilon}\right)}{n}} + \sqrt{\epsilon} \quad\quad (22)
\end{aligned}
$$

We take $\epsilon = \sqrt{\frac{W_eL_e(\log W_e + L_e)\log(n\ell)}{n}}$ makes $\mathcal{R}((\Phi\circ\mathcal{E}), X_{1:n}) \lesssim \sqrt{\frac{W_eL_e(\log W_e + L_e)\log(n\ell)}{n}}$. $\square$

### C.4.6 PROOF OF LEMMA 25

To prove Lemma 25, we need some supporting results, which we sequentially state and prove as follows. The first such result, i.e. Lemma 27 ensures that the kernel function is Lipschitz when it is considered as a map from a real vector space to the corresponding Hilbert space.

**Lemma 27.** *Suppose assumption A2 holds. Then,* $\|\mathcal{K}(x,\cdot) - \mathcal{K}(y,\cdot)\|_{\mathbb{H}_\mathcal{K}}^2 \leq 2\tau_k\|x - y\|_2$.

*Proof.* We observe the following:

$$
\begin{aligned}
\|\mathcal{K}(x,\cdot) - \mathcal{K}(y,\cdot)\|_{\mathbb{H}_\mathcal{K}}^2 =& \mathcal{K}(x,x) + \mathcal{K}(y,y) - 2\mathcal{K}(x,y) \\
=& (\mathcal{K}(x,x) - \mathcal{K}(x,y)) + (\mathcal{K}(y,y) - \mathcal{K}(x,y)) \\
\leq& 2\tau_k\|x - y\|_2.
\end{aligned}
$$

$\square$

Lemma 28 states that the difference between the estimated and actual squared MMD-dissimilarity scales as $\mathcal{O}(1/n)$ for estimates (5) and $\mathcal{O}(1/n + 1/m)$ for estimates (6).

**Lemma 28.** *Suppose assumption A2 holds. Then, for any $E \in \mathcal{E}$,*

(a) $\left|\widehat{MMD}_\mathcal{K}^2(E_\sharp\hat{\mu}_n, \nu) - MMD_\mathcal{K}^2(E_\sharp\hat{\mu}_n, \nu)\right| \leq \frac{2B^2}{n}$.

(b) $\left|\widehat{MMD}_\mathcal{K}^2(E_\sharp\hat{\mu}_n, \hat{\nu}_m) - MMD_\mathcal{K}^2(E_\sharp\hat{\mu}_n, \nu)\right| \leq 2B^2\left(\frac{1}{n} + \frac{1}{m}\right)$.

*Proof.* We note that,

$$\widehat{\text{MMD}}^2_{\mathcal{K}}(E_\sharp\hat{\mu}_n, \nu) - \text{MMD}^2(E_\sharp\hat{\mu}_n, \nu)$$

$$= \frac{1}{n(n-1)} \sum_{i \neq j} \mathcal{K}(E(X_i), E(X_j)) - \frac{1}{n^2} \sum_{i,j=1}^{n} \mathcal{K}(E(X_i), E(X_j))$$

$$= \frac{1}{n^2(n-1)} \sum_{i \neq j} \mathcal{K}(E(X_i), E(X_j)) - \frac{1}{n^2} \sum_{i=1}^{n} \mathcal{K}(E(X_i), E(X_i))$$

Thus,

$$\left| \widehat{\text{MMD}}^2_{\mathcal{K}}(E_\sharp\hat{\mu}_n, \nu) - \text{MMD}^2(E_\sharp\hat{\mu}_n, \nu) \right| \leq \frac{1}{n^2(n-1)} \times n(n-1)B^2 + \frac{1}{n^2} \times nB^2 = \frac{2B^2}{n}.$$

Part (b) follows similarly. $\square$

We also note that the $\text{MMD}_{\mathcal{K}}$-metric is bounded under A2 as seen in Lemma 29.

**Lemma 29.** *Under assumption A2, $MMD_{\mathcal{K}}(P, Q) \leq 2B$, for any two distributions $P$ and $Q$.*

*Proof.* $|f(x)| = \langle \mathcal{K}(x, \cdot), f \rangle \leq \|\mathcal{K}(x, \cdot)\|_{\mathbb{H}_{\mathcal{K}}} = B$. This implies that $\text{MMD}_{\mathcal{K}}(P, Q) = \sup_{\phi \in \Phi}(\int \phi dP - \int \phi dQ) \leq 2B$ $\square$

**Lemma 30.** *Suppose assumption A2 holds. Then,*

*(a)* $\mathbb{E} \sup_{E \in \mathcal{E}} \left| \widehat{MMD}_{\mathcal{K}}(E_\sharp\hat{\mu}_n, \nu) - MMD_{\mathcal{K}}(E_\sharp\mu, \nu) \right| \lesssim \sqrt{\frac{W_e L_e (\log W_e + L_e) \log(n\ell)}{n}}.$

*(b)* $\mathbb{E} \sup_{E \in \mathcal{E}} \left| \widehat{MMD}_{\mathcal{K}}(E_\sharp\hat{\mu}_n, \hat{\nu}_m) - MMD_{\mathcal{K}}(E_\sharp\mu, \nu) \right| \lesssim \sqrt{\frac{W_e L_e (\log W_e + L_e) \log(n\ell)}{n}} + \frac{1}{\sqrt{m}}.$

*Proof.* **Proof of Part (a)**

We begin by noting that,

$$\mathbb{E} \sup_{E \in \mathcal{E}} \left| \widehat{\text{MMD}}_{\mathcal{K}}(E_\sharp\hat{\mu}_n, \nu) - \text{MMD}_{\mathcal{K}}(E_\sharp\mu, \nu) \right|$$

$$\leq \mathbb{E} \sup_{E \in \mathcal{E}} |\text{MMD}_{\mathcal{K}}(E_\sharp\hat{\mu}_n, \nu) - \text{MMD}_{\mathcal{K}}(E_\sharp\mu, \nu)| + \mathbb{E} \sup_{E \in \mathcal{E}} \left| \widehat{\text{MMD}}_{\mathcal{K}}(E_\sharp\hat{\mu}_n, \nu) - \text{MMD}_{\mathcal{K}}(E_\sharp\hat{\mu}_n, \nu) \right|$$

$$\leq \mathbb{E} \sup_{E \in \mathcal{E}} \text{MMD}_{\mathcal{K}}(E_\sharp\hat{\mu}_n, E_\sharp\mu) + 2B\sqrt{\frac{1}{n}} \tag{23}$$

$$= \mathbb{E} \sup_{E \in \mathcal{E}} \sup_{\phi \in \Phi} \left( \int \phi(E(x)) d\hat{\mu}_n(x) - \int \phi(E(x)) d\mu(x) \right) + 2B\sqrt{\frac{1}{n}}$$

$$\leq 2\mathcal{R}(\Phi \circ \mathcal{E}, \mu) + 2B\sqrt{\frac{1}{n}} \tag{24}$$

$$\lesssim \sqrt{\frac{W_e L_e (\log W_e + L_e) \log(n\ell)}{n}}. \tag{25}$$

In the above calculations, (23) follows from Lemma 28. Inequality (24) follows from symmetrization, whereas, (25) follows from Lemma 24.

**Proof of Part (b)** Similar to the calculations in part (a), we note the following:

$$\mathbb{E} \sup_{E \in \mathcal{E}} \left| \widehat{\text{MMD}}_{\mathcal{K}}(E_\sharp\hat{\mu}_n, \hat{\nu}_m) - \text{MMD}_{\mathcal{K}}(E_\sharp\mu, \nu) \right|$$

$$\leq \mathbb{E} \sup_{E \in \mathcal{E}} |\text{MMD}_{\mathcal{K}}(E_\sharp\hat{\mu}_n, \hat{\nu}_m) - \text{MMD}_{\mathcal{K}}(E_\sharp\mu, \nu)| + \mathbb{E} \sup_{E \in \mathcal{E}} \left| \widehat{\text{MMD}}_{\mathcal{K}}(E_\sharp\hat{\mu}_n, \hat{\nu}_m) - \text{MMD}_{\mathcal{K}}(E_\sharp\hat{\mu}_n, \nu) \right|$$

$$\leq \mathbb{E} \sup_{E \in \mathcal{E}} \text{MMD}_{\mathcal{K}}(E_\sharp\hat{\mu}_n, E_\sharp\mu) + \mathbb{E}\text{MMD}_{\mathcal{K}}(\hat{\nu}_m, \nu) + 2B\sqrt{\frac{1}{n} + \frac{1}{m}}$$

$$\leq 2\mathcal{R}(\Phi \circ \mathcal{E}, \mu) + \mathcal{R}(\Phi, \nu) + 2B\sqrt{\frac{1}{n} + \frac{1}{m}}$$

$$\lesssim \sqrt{\frac{W_e L_e (\log W_e + L_e) \log(n\ell)}{n}} + \frac{1}{\sqrt{m}}.$$

$\square$

We are now ready to prove Lemma 25. For ease of readability, we restate the Lemma as follows.

**Lemma 25.** *Under assumption A2, the following holds:*

*(a)* $\mathbb{E} \sup\limits_{E \in \mathcal{E}} \left| \widehat{MMD}^2_{\mathcal{K}}(E_\sharp \hat{\mu}_n, \nu) - MMD^2_{\mathcal{K}}(E_\sharp \mu, \nu) \right| \lesssim \sqrt{\frac{W_e L_e \log W_e \log(n\ell)}{n}},$

*(b)* $\mathbb{E} \sup\limits_{E \in \mathcal{E}} \left| \widehat{MMD}^2_{\mathcal{K}}(E_\sharp \hat{\mu}_n, \hat{\nu}_m) - MMD^2_{\mathcal{K}}(E_\sharp \mu, \nu) \right| \lesssim \sqrt{\frac{W_e L_e (\log W_e + L_e) \log(n\ell)}{n}} + \frac{1}{\sqrt{m}}.$

*Proof.* **Proof of part (a)** We begin by noting the following:

$$\mathbb{E} \sup_{E \in \mathcal{E}} \left| \widehat{MMD}^2_{\mathcal{K}}(E_\sharp \hat{\mu}_n, \nu) - \mathrm{MMD}^2_{\mathcal{K}}(E_\sharp \mu, \nu) \right|$$

$$= \mathbb{E} \sup_{E \in \mathcal{E}} \left| 2\mathrm{MMD}_{\mathcal{K}}(E_\sharp \mu, \nu) \left( \widehat{\mathrm{MMD}}_{\mathcal{K}}(E_\sharp \hat{\mu}_n, \nu) - \mathrm{MMD}_{\mathcal{K}}(E_\sharp \mu, \nu) \right) + \left( \widehat{\mathrm{MMD}}_{\mathcal{K}}(E_\sharp \hat{\mu}_n, \nu) - \mathrm{MMD}_{\mathcal{K}}(E_\sharp \mu, \nu) \right)^2 \right|$$

$$\leq 2B \mathbb{E} \sup_{E \in \mathcal{E}} \left| \widehat{\mathrm{MMD}}_{\mathcal{K}}(E_\sharp \hat{\mu}_n, \nu) - \mathrm{MMD}_{\mathcal{K}}(E_\sharp \mu, \nu) \right| + \mathbb{E} \sup_{E \in \mathcal{E}} \left| \widehat{\mathrm{MMD}}_{\mathcal{K}}(E_\sharp \hat{\mu}_n, \nu) - \mathrm{MMD}_{\mathcal{K}}(E_\sharp \mu, \nu) \right|^2 \tag{26}$$

$$\lesssim \sqrt{\frac{\ell W_e L_e (\log W_e + L_e) \log n}{n}}. \tag{27}$$

Inequality (26) follows from applying Lemma 29, whereas, (27) is a consequence of Lemma (30).

**Proof of part (b)** Similarly,

$$\mathbb{E} \sup_{E \in \mathcal{E}} \left| \widehat{MMD}^2_{\mathcal{K}}(E_\sharp \hat{\mu}_n, \hat{\nu}_m) - \mathrm{MMD}^2_{\mathcal{K}}(E_\sharp \mu, \nu) \right|$$

$$= \mathbb{E} \sup_{E \in \mathcal{E}} \left| 2\mathrm{MMD}_{\mathcal{K}}(E_\sharp \mu, \nu) \left( \widehat{\mathrm{MMD}}_{\mathcal{K}}(E_\sharp \hat{\mu}_n, \hat{\nu}_m) - \mathrm{MMD}_{\mathcal{K}}(E_\sharp \mu, \nu) \right) + \left( \widehat{\mathrm{MMD}}_{\mathcal{K}}(E_\sharp \hat{\mu}_n, \hat{\nu}_m) - \mathrm{MMD}_{\mathcal{K}}(E_\sharp \mu, \nu) \right)^2 \right|$$

$$\leq 2B \mathbb{E} \sup_{E \in \mathcal{E}} \left| \widehat{\mathrm{MMD}}_{\mathcal{K}}(E_\sharp \hat{\mu}_n, \hat{\nu}_m) - \mathrm{MMD}_{\mathcal{K}}(E_\sharp \mu, \nu) \right| + \mathbb{E} \sup_{E \in \mathcal{E}} \left| \widehat{\mathrm{MMD}}_{\mathcal{K}}(E_\sharp \hat{\mu}_n, \hat{\nu}_m) - \mathrm{MMD}_{\mathcal{K}}(E_\sharp \mu, \nu) \right|^2$$

$$\lesssim \sqrt{\frac{\ell W_e L_e (\log W_e + L_e) \log n}{n}} + \frac{1}{\sqrt{m}}.$$

$\square$

## C.5 PROOFS FROM SECTION 5.2

In this section, we prove the main result of this paper, i.e. Theorem 8.

## C.6 PROOFS FROM SECTION 5.5

To begin our analysis, we first show the following:

**Theorem 31.** *Under assumptions, A1–3, $V(\mu, \nu, \hat{G}_n, \hat{E}) \to 0$, almost surely.*

*Proof.* For simplicity, we consider the estimator (5). A similar proof holds for estimator (6). Consider the oracle inequality (7). We only consider the case, when, diss $= \mathcal{W}_1$, the case when, diss $= \mathrm{MMD}^2_{\mathcal{K}}$ can be proved similarly.

We note that $\mathcal{F}$ is a bounded function class, with bound $B_c$. Thus, a simple application of the bounded difference inequality yields that with probability at least $1 - \delta/2$,

$$\|\hat{\mu}_n - \mu\|_{\mathcal{F}} \leq \mathbb{E}\|\hat{\mu}_n - \mu\|_{\mathcal{F}} + \theta_1 \sqrt{\frac{\log(1/\delta)}{n}},$$

for some positive constant $\theta_1$. The fourth term in (5) can be written as:

$$\sup_{E \in \mathcal{E}} |\mathcal{W}_1(E_\sharp \hat{\mu}_n, \nu) - \mathcal{W}_1(E_\sharp \mu, \nu)|.$$

Suppose that $\hat{\mu}'_n$ denotes the empirical distribution on $(X_1, \ldots, X_{i-1}, X'_i, \ldots, X_n)$. Then replacing $\hat{\mu}_n$ with $\hat{\mu}'_n$, yields an error at most,

$$\sup_{E \in \mathcal{E}} |\mathcal{W}_1(E_\sharp \hat{\mu}_n, \nu) - \mathcal{W}_1(E_\sharp \hat{\mu}'_n, \nu)| \leq \sup_{E \in \mathcal{E}} \mathcal{W}_1(E_\sharp \hat{\mu}_n, E_\sharp \hat{\mu}'_n) \leq \sup_{E \in \mathcal{E}} \sup_{f : \|f\|_{\text{Lip}} \leq 1} \frac{1}{n} |f(E(X_i)) - f(E(X'_i))| \lesssim \frac{1}{n},$$

since by construction, $E$'s are chosen from bounded ReLU functions. Again by a simple application of bounded difference inequality, we get,

$$2\lambda \sup_{E \in \mathcal{E}} |\mathcal{W}_1(E_\sharp \hat{\mu}_n, \nu) - \mathcal{W}_1(E_\sharp \mu, \nu)| \leq 2\lambda \mathbb{E} \sup_{E \in \mathcal{E}} |\mathcal{W}_1(E_\sharp \hat{\mu}_n, \nu) - \mathcal{W}_1(E_\sharp \mu, \nu)| + \theta_2 \sqrt{\frac{\log(1/\delta)}{n}},$$

with probability at least $1 - \delta/2$. Hence, by union bound, with probability at least $1 - \delta$

$$2\|\hat{\mu}_n - \mu\|_{\mathcal{F}} + 2\lambda \sup_{E \in \mathcal{E}} |\mathcal{W}_1(E_\sharp \hat{\mu}_n, \nu) - \mathcal{W}_1(E_\sharp \mu, \nu)|$$

$$\leq 2\mathbb{E}\|\hat{\mu}_n - \mu\|_{\mathcal{F}} + 2\lambda \mathbb{E} \sup_{E \in \mathcal{E}} |\mathcal{W}_1(E_\sharp \hat{\mu}_n, \nu) - \mathcal{W}_1(E_\sharp \mu, \nu)| + \theta_3 \sqrt{\frac{\log(1/\delta)}{n}},$$

for some absolute constant $\theta_3$. Since, all other terms in (7) are bounded independent of the random sample, with probability at least $1 - \delta$,

$$V(\mu, \nu, \hat{G}, \hat{E}) \leq \mathbb{E}V(\mu, \nu, \hat{G}, \hat{E}) + \theta_3 \sqrt{\frac{\log(1/\delta)}{n}}.$$

From the above, $\mathbb{P}(|V(\mu, \nu, \hat{G}, \hat{E}) - \mathbb{E}V(\mu, \nu, \hat{G}, \hat{E})| > \epsilon) \leq e^{-\frac{n\epsilon^2}{\theta_3}}$. this implies that $\sum_{n \geq 1} \mathbb{P}(|V(\mu, \nu, \hat{G}, \hat{E}) - \mathbb{E}V(\mu, \nu, \hat{G}, \hat{E})| > \epsilon) < \infty$. A simple application of the first Borel-Cantelli Lemma yields (see Proposition 5.7 of Karr (1993)) that this implies that $|V(\mu, \nu, \hat{G}, \hat{E}) - \mathbb{E}V(\mu, \nu, \hat{G}, \hat{E})| \to 0$, almost surely. Since, $\lim_{n \to \infty} \mathbb{E}V(\mu, \nu, \hat{G}, \hat{E}) = 0$, the result follows. $\qquad\square$

### C.6.1 PROOF OF PROPOSITION 12

**Proposition 12.** *Suppose that assumptions A1–3 hold. Then, for both the dissimilarity measures $\mathcal{W}_1(\cdot, \cdot)$ and $MMD^2_{\mathcal{K}}(\cdot, \cdot)$ and the estimates (5) and (6), $\hat{E}_\sharp \mu \xrightarrow{d} \nu$, almost surely.*

*Proof.* Let diss $= \mathcal{W}_1$. From Theorem 31, it is clear that, $\mathcal{W}_1(\hat{E}_\sharp \mu, \nu) \to 0$, almost surely. Since convergence in Wasserstein distance characterizes convergence in distribution, $\hat{E}_\sharp \mu \xrightarrow{d} \nu$, almost surely.

When, diss $= MMD^2_{\mathcal{K}}$, we can similarly say that $MMD^2_{\mathcal{K}}(\hat{E}_\sharp \mu, \nu) \to 0$, almost surely. From Theorem 3.2 (b) of Schreuder et al. (2021), we conclude that $\hat{E}_\sharp \mu \xrightarrow{d} \nu$, almost surely. $\qquad\square$

### C.6.2 PROOF OF PROPOSITION 13

**Proposition 13.** *Suppose that assumptions A1–3 hold. Then, for both the dissimilarity measures $\mathcal{W}_1(\cdot, \cdot)$ and $MMD^2_{\mathcal{K}}(\cdot, \cdot)$ and the estimates (5) and (6), $\|id(\cdot) - \hat{G} \circ \hat{E}(\cdot)\|^2_{\mathbb{L}_2(\mu)} \xrightarrow{a.s.} 0$.*

*Proof.* The proof follows from observing that $0 \leq \|id(\cdot) - \hat{G} \circ \hat{E}(\cdot)\|^2_{\mathbb{L}_2(\mu)} \leq V(\mu, \nu, \hat{G}, \hat{E})$ and applying Theorem 31. $\qquad\square$

### C.6.3 PROOF OF THEOREM 14

**Theorem 14.** *Suppose that assumptions A1–3 hold and $TV(\hat{E}_\sharp \mu, \nu) \to 0$, almost surely. Then, $\hat{G}_\sharp \nu \xrightarrow{d} \mu$, almost surely.*

*Proof.* We begin by observing that,

$$\mathcal{W}_1(\hat{G}_\sharp \nu, \mu) \leq \mathcal{W}_1(\hat{G}_\sharp \nu, (\hat{G} \circ \hat{E})_\sharp \mu) + \mathcal{W}_1((\hat{G} \circ \hat{E})_\sharp \mu, \mu) \tag{28}$$

We first note that

$$
\begin{aligned}
TV(\hat{G}_\sharp \nu, (\hat{G} \circ \hat{E})_\sharp \mu) &= \sup_{B \in \mathcal{B}(\mathbb{R}^d)} |(\hat{G}_\sharp \nu)(B) - ((\hat{G} \circ \hat{E})_\sharp \mu)(B)| \\
&= \sup_{B \in \mathcal{B}(\mathbb{R}^d)} |\nu(\hat{G}^{-1}(B)) - (\hat{E}_\sharp \mu)(\hat{G}^{-1}(B))| \\
&\leq \sup_{B \in \mathcal{B}(\mathbb{R}^\ell)} |\nu(B) - (\hat{E}_\sharp \mu)(B)| \tag{29} \\
&= TV(\nu, \hat{E}_\sharp \mu) \to 0, \text{ almost surely.}
\end{aligned}
$$

Here (29) follows from the fact that $\left\{ \hat{G}^{-1}(B) : B \in \mathbb{R}^d \right\} \subseteq \mathbb{R}^\ell$, since $\hat{G}$'s are measurable. Thus, $TV(\hat{G}_\sharp \nu, (\hat{G} \circ \hat{E})_\sharp \mu) \to 0$, almost surely. Since convergence in TV implies convergence in distribution, this implies that $\mathcal{W}_1(\hat{G}_\sharp \nu, (\hat{G} \circ \hat{E})_\sharp \mu) \to 0$, almost surely.

We also note that, from Proposition 13, $\mathbb{E}_{X \sim \mu} \|X - \hat{G} \circ \hat{E}(X)\|^2 \to 0$, almost surely. This implies that $\|X - \hat{G} \circ \hat{E}(X)\| \xrightarrow{P} 0$, almost surely, which further implies that $\hat{G} \circ \hat{E}(X) \xrightarrow{d} X$, almost surely. Hence, $\mathcal{W}_1((\hat{G} \circ \hat{E})_\sharp \mu, \mu) \to 0$, almost surely. Plugging these in (28) gives us the desired result. $\qquad\square$

**Theorem 15.** *Suppose that assumptions A1–3 hold and let the family of estimated generators $\{\hat{G}^n\}_{n \in \mathbb{N}}$ be uniformly equicontinuous, almost surely. Then, $\hat{G}_\sharp^n \nu \xrightarrow{d} \mu$, almost surely.*

*Proof.* We note that from the proof of Theorem 14, equation (28) holds and $\mathcal{W}_1((\hat{G}^n \circ \hat{E}^n)_\sharp \mu, \mu) \to 0$, almost surely. We fix an $\omega$ in the sample space, for which, $\mathcal{W}_1((\hat{G}_\omega^n \circ \hat{E}_\omega^n)_\sharp \mu, \mu) \to 0$ and $(\hat{E}_\omega^n)_\sharp \mu \xrightarrow{d} \nu$. Here we use the subscript $\omega$ to show that $\hat{G}^n$ and $\hat{E}^n$ might depend on $\omega$. Clearly, the set of all $\omega$'s, for which this convergence holds, has probability 1.

By Skorohod's theorem, we note that we can find a sequence of random variables $\{Y_n\}_{n \in \mathbb{N}}$ and $Z$, such that $Y_n$ follows the distribution $\hat{E}_\sharp^n \mu$ and $Z \sim \nu$, such that $Y_n \xrightarrow{a.s.} Z$. Since $\{\hat{G}_\omega^n\}_{n \in \mathbb{N}}$ are uniformly equicontinuous, for any $\epsilon > 0$, we can find $\delta > 0$, such that if $|y_n - z| < \delta$, $|\hat{G}_\omega^n(y_n) - \hat{G}_\omega^n(z)| < \epsilon$. Thus, $\hat{G}_\omega^n(Y_n) - \hat{G}_\omega^n(Z) \xrightarrow{a.s.} 0$. Since this implies that $\hat{G}_\omega^n(Y_n) - \hat{G}_\omega^n(Z) \xrightarrow{d} 0$, it is easy to see that, $\mathcal{W}_1(\hat{G}_\omega^n(Y_n), \hat{G}_\omega^n(Z)) \to 0$. Now, since, $\mathcal{W}_1(\hat{G}_\omega^n(Y_n), \hat{G}_\omega^n(Z)) = \mathcal{W}_1((\hat{G}_\omega^n)_\sharp \nu, (\hat{G}_\omega^n \circ \hat{E}_\omega^n)_\sharp \mu)$, we conclude that $\mathcal{W}_1((\hat{G}_\omega^n)_\sharp \nu, (\hat{G}_\omega^n \circ \hat{E}_\omega^n)_\sharp \mu) \to 0$, as $n \to \infty$. Thus, with probability one, the RHS of (28) goes to 0 as $n \to \infty$. Hence, $\mathcal{W}_1(\hat{G}_\sharp^n \nu, \mu) \to 0$, almost surely. $\qquad\square$

### C.6.4 PROOF OF COROLLARY 16

**Corollary 16.** *Let $diss(\cdot, \cdot) = \mathcal{W}_1(\cdot, \cdot)$ and suppose that the assumptions of Theorem 8 are satisfied and $s > d_\mu$. Also let $\sup_{n \in \mathbb{N}} \|\hat{G}^n\|_{Lip}, \sup_{m,n \in \mathbb{N}} \|\hat{G}^{n,m}\|_{Lip} \leq L$, almost surely, for some $L > 0$. $\mathcal{W}_1(\hat{G}_\sharp \nu, \mu) \lesssim V(\mu, \nu, \hat{G}, \hat{E})$ for both estimators (5) and (6).*

*Proof.* Denoting $\hat{G}$ as either of the estimators (5) and (6), it is easy to see that,

$$
\begin{aligned}
\mathcal{W}_1(\hat{G}_\sharp \nu, \mu) &\leq \mathcal{W}_1(\hat{G}_\sharp \nu, (\hat{G} \circ \hat{E})_\sharp \mu) + \mathcal{W}_1((\hat{G} \circ \hat{E})_\sharp \mu, \mu) \\
&\leq L \mathcal{W}_1(\nu, \hat{E}_\sharp \mu) + \mathcal{W}_1((\hat{G} \circ \hat{E})_\sharp \mu, \mu)
\end{aligned}
$$

$$\lesssim \mathcal{W}_1(\nu, \hat{E}_\sharp \mu) + \int \|\hat{G} \circ \hat{E}(x) - x\|_2^2 d\mu(x)$$

$\square$

## D  SUPPORTING RESULTS FOR APPROXIMATION GUARANTEES

**Lemma 32.** *(Proposition 2 of Yarotsky (2017)) The function $f(x) = x^2$ on the segment $[0,1]$ can be approximated with any error by a ReLU network, $sq_m(\cdot)$, such that,*

1. $\mathcal{L}(sq_m), \mathcal{W}(sq_m) \leq c_1 m$.

2. $sq_m\left(\frac{k}{2^m}\right) = \left(\frac{k}{2^m}\right)^2$, *for all* $k = 0, 1, \ldots, 2^m$.

3. $\|sq_m - x^2\|_{\mathbb{L}_\infty([0,1])} \leq \frac{1}{2^{2m+2}}$.

**Lemma 33.** *Let $sq_m(\cdot)$ be taken as in Lemma 32, then, $\|sq_m - x^2\|_{\mathcal{H}^\beta} \leq \frac{1}{2^{m-1}}$.*

*Proof.* We begin by noting that, $\mathrm{sq}_m(x) = \left(\frac{(k+1)^2}{2^m} - \frac{k^2}{2^m}\right)\left(x - \frac{k}{2^m}\right) + \left(\frac{k}{2^m}\right)^2$, whenever, $x \in \left[\frac{k}{2^m}, \frac{k+1}{2^m}\right)$. Thus, on $\left(\frac{k}{2^m}, \frac{k+1}{2^m}\right)$),

$$\|\mathrm{sq}_m - x^2\|_{\mathcal{H}^\beta} = \|\mathrm{sq}_m - x^2\|_{\mathbb{L}_\infty\left(\left(\frac{k}{2^m}, \frac{k+1}{2^m}\right)\right)} + \left\|\frac{(k+1)^2}{2^m} - \frac{k^2}{2^m} - 2x\right\|_{\mathbb{L}_\infty\left(\left(\frac{k}{2^m}, \frac{k+1}{2^m}\right)\right)} = \frac{1}{2^{2m+2}} + \frac{1}{2^m} \leq \frac{1}{2^{m-1}}.$$

This implies that, $\|\mathrm{sq}_m - x^2\|_{\mathcal{H}^\beta} \leq \frac{1}{2^{m-1}}$. $\square$

**Lemma 34.** *Let $M > 0$, then we can find a ReLU network $prod_m^{(2)}$, such that,*

1. $\mathcal{L}(prod_m^{(2)}), \mathcal{W}(prod_m^{(2)}) \leq c_2 m$, *for some absolute constant $c_2$.*

2. $\|prod_m^{(2)} - xy\|_{\mathbb{L}_\infty([-M,M]\times[-M,M])} \leq \frac{M^2}{2^{2m+1}}$.

*Proof.* Let $\mathrm{prod}_m^{(2)}(x,y) = M^2\left(\mathrm{sq}_m\left(\frac{|x+y|}{2M}\right) - \mathrm{sq}_m\left(\frac{|x-y|}{2M}\right)\right)$. Clearly, $\mathrm{prod}_m^{(2)}(x,y) = 0$, if $xy = 0$. We note that, $\mathcal{L}(\mathrm{prod}_m^{(2)}) \leq c_1 m + 1 \leq c_2 m$ and $\mathcal{W}(\mathrm{prod}_m^{(2)}) \leq 2c_1 m + 2 \leq c_2 m$, for some absolute constant $c_2$. Clearly,

$$\|\mathrm{prod}_m^{(2)} - xy\|_{\mathbb{L}_\infty([-M,M]\times[-M,M])} \leq 2M^2\|\mathrm{sq} - x^2\|_{\mathbb{L}([0,1])} \leq \frac{M^2}{2^{2m+1}}.$$

$\square$

**Lemma 35.** *For any $m \geq 3$, we can construct a ReLU network $prod_m^{(d)} : \mathbb{R}^d \to \mathbb{R}$, such that for any $x_1, \ldots, x_d \in [-1,1]$, $\|prod_m^{(d)}(x_1, \ldots, x_d) - x_1 \ldots x_d\|_{\mathbb{L}_\infty([-1,1]^d)} \leq \frac{d^3}{2^{2m+2}}$.*

*Proof.* Let $M = 1$ and $d \geq 2$. We define $\mathrm{prod}_m^{(k)}(x_1, \ldots, x_k) = \mathrm{prod}_m^{(2)}(\mathrm{prod}_m^{(k-1)}(x_1, \ldots, x_{k-1}), x_d)$, $k \geq 3$. Clearly $\mathcal{W}(\mathrm{prod}_m^{(d)}), \mathcal{L}(\mathrm{prod}_m^{(d)}) \leq c_3 dm$, for some absolute constant $c_3$. We also note that, $|\mathrm{prod}_m^{(d)}(x_1, \ldots, x_d)| \leq \frac{M^2}{2^{2m+1}} + x_d|\mathrm{prod}_m^{(d-1)}(x_1, \ldots, x_{d-1})| \leq \frac{M^2}{2^{2m+1}} + M|\mathrm{prod}_m^{(d-1)}(x_1, \ldots, x_{d-1})| \leq \frac{M^2}{2^{2m+1}} + \frac{M^3}{2^{2m+1}} + \cdots + \frac{M^{d-1}}{2^{2m+1}} + M^d \leq \frac{M^2}{2^{2m+1}} + (d-2)M^d = d - 2 + \frac{1}{2^{2m+1}} \leq d - 1$. From induction, it is easy to see that, $\mathrm{prod}_m^{(k)} \leq d - 1$. Taking $M = d - 1$, we get that,

$$\|\mathrm{prod}_m^{(d)}(x_1, \ldots, x_d) - x_1 \ldots x_d\|_{\mathbb{L}_\infty([-1,1]^d)}$$
$$= \|\mathrm{prod}_m^{(2)}(\mathrm{prod}_m^{(d-1)}(x_1, \ldots, x_{d-1}), x_d) - x_1 \ldots x_d\|_{\mathbb{L}_\infty([-1,1]^d)}$$
$$\leq \|\mathrm{prod}_m^{(d-1)}(x_1, \ldots, x_{d-1}) - x_1 \ldots x_{d-1}\|_{\mathbb{L}_\infty([-1,1]^d)} + \frac{M^2}{2^{2m+2}}$$

$$\leq \frac{dM^2}{2^{2m+2}}$$
$$= \frac{d^3}{2^{2m+2}}.$$

$\square$

# E    SUPPORTING RESULTS FROM THE LITERATURE

This section lists some of the supporting results from the literature, used in the paper.

**Lemma 36.** *(Kolmogorov & Tikhomirov, 1961) The $\epsilon$-covering number of $\mathcal{H}^\beta([0,1]^d, \mathbb{R}, 1)$ can be bounded as,*
$$\log \mathcal{N}\left((\epsilon; \mathcal{H}^\beta([0,1]^d), \|\cdot\|_\infty\right) \lesssim \epsilon^{-d/\beta}.$$

**Lemma 37.** *(Theorem 12.2 of Anthony & Bartlett (2009)) Assume for all $f \in \mathcal{F}$, $\|f\|_\infty \leq M$. Denote the pseudo-dimension of $\mathcal{F}$ as $Pdim(\mathcal{F})$, then for $n \geq Pdim(\mathcal{F})$, we have for any $\epsilon$ and any $X_1, \ldots, X_n$,*
$$\mathcal{N}\epsilon, \mathcal{F}_{|X_{1:n}}, \ell_\infty) \leq \left(\frac{2eMn}{\epsilon Pdim(\mathcal{F})}\right)^{Pdim(\mathcal{F})}.$$

**Lemma 38.** *(Theorem 6 of Bartlett et al. (2019)) Consider the function class computed by a feed-forward neural network architecture with $W$ many weight parameters and $U$ many computation units arranged in $L$ layers. Suppose that all non-output units have piecewise-polynomial activation functions with $p+1$ pieces and degrees no more than $d$, and the output unit has the identity function as its activation function. Then the VC-dimension and pseudo-dimension are upper-bounded as*

$$VCdim(\mathcal{F}), Pdim(\mathcal{F}) \leq C \cdot LW \log(pU) + L^2 W \log d.$$

