# OpenReview forum: "A Statistical Analysis of Wasserstein Autoencoders for Intrinsically Low-dimensional Data"
_ICLR.cc/2024/Conference — ICLR 2024 poster_

### Official Review · Reviewer_e5jX · 2023-10-28

**Soundness:** 3 good
**Presentation:** 3 good
**Contribution:** 3 good
**Rating:** 6
**Confidence:** 3

**Summary:**

The paper summarizes the statistical error bounds for Wasserstein Autoencoders (WAE). Assuming the existence of the true smooth autoencoder model and (intrinsic) low-dimensionality of data distribution, this paper derives the order of the errors of the estimated model with respect to sample number $n$ and other assumed quantities. The resulting corollaries are also given.

**Strengths:**

The asymptotical error bounds for the estimated WAE is clearly presented with thorough proofs. This guarantess the asymptotical convergence of the WAE model if the target distribution and the true encoder/decoder are well-regularized.

**Weaknesses:**

The assumption of the existence of true generator and the true encoder seems to be a strong assumption. Practically, the low-dimensional prior distribution is chosen to be Gaussian, and this distribution may not be able to represent the intrinsic structure of the data distribution.

Also, it is common to use GAN loss as the dissimilarity measure for the penalty in the objective function, but it seems that GAN loss is not considered.

**Questions:**

Can you show a numerical experiment that supports up the asymptotical rate of the presented error bound? Figure 1 may be close to this, but the comparison with $O(n^\alpha)$ or quantity of the whole objective function should be presented.

Also, is the theorem applicable for WAE-GAN?

---

> ### Author Response · Authors · 2023-11-21
> **Response to Reviewer e5jX**
>
> We thank the reviewer for their overall positive reception of our work and insightful comments. We are pleased to note that you found the asymptotic error bounds for the estimated WAE to be clearly presented with thorough proofs. We agree that these bounds are crucial in guaranteeing the asymptotical convergence of the WAE model, provided the assumptions about the target distribution and true encoder/decoder regularization hold.
>
>  We appreciate the constructive criticism and insightful comments provided by the reviewer. We agree with the reviewer that assumption A1 is quite strong as highlighted in the paragraph following A1 on page 6 of the manuscript. Certainly, in the context of GANs, a less stringent assumption can be employed since the learning task becomes considerably simpler. A parallel challenge arose in the analysis of autoencoders, as elucidated by Liu et al. (2023). They addressed this issue by exploring chart-autoencoders, which incorporate additional components in the network architecture compared to conventional autoencoders. As already observed by the reviewer, the empirical evidence presented in Section 2 illustrates a polynomial-like decay pattern as hinted at by Theorem 8 and its corollaries. Experiments on different other classes of Imagenet suggest a similar trend showing roughly a polynomial decay. In response to the reviewer's recommendation, we will broaden the experimental study to encompass additional datasets.
>
>  Though the main theorem is not directly applicable to GAN-WAE, one can think of the case when the dissimilarity measure is taken to be the Wasserstein-1 distance to be a close replica of GAN-WAE. This is because when the discriminator is large, one can expect that after weight-clipping, any Lipschitz function can be closely realized by them. Thus, when optimizing on the discriminator, one can assume that the supremum is roughly attained (as the discriminator is assumed to be a large network). However, when working with a fixed class of discriminators, one can possibly prove analogous theorems in the so-called neural network distance. Exploring this avenue further would be an intriguing direction for future study. We appreciate the reviewer for raising this thought-provoking question and will incorporate this insightful discussion into the revised manuscript.

---

> ### Comment · Reviewer_e5jX · 2023-11-22
>
> I thank the authors for their kind replies. My questions are all responded, and I will retain my scores.

---

### Official Review · Reviewer_JGBa · 2023-11-01

**Soundness:** 4 excellent
**Presentation:** 3 good
**Contribution:** 3 good
**Rating:** 6
**Confidence:** 4

**Summary:**

The paper proves statistical sample complexity for Wasserstein autoencoders on low-dimensional data. The provided sample complexity depends on the data intrinsic dimension, indicating that autoencoders are adaptive to the data intrinsic structures. The analysis also includes MMD divergence. This study is relatively new in the literature, as it is parallel to the GANs analyses and compared to "Deep nonparametric estimation of intrinsic data structures by chart autoencoders: Generalization error and robustness", this paper includes the regularization using Wasserstein distance or MMD.

**Strengths:**

The paper is quite theoretical, but the presentation is relatively easy to follow. The error decomposition in Lemma 7 is helpful in understanding the rate in later sections. As far as I can tell, the theoretical claims are correct.

The paper also provides some numerical results suggesting that the performance of autoencoders depends on data intrinsic dimension. This is good to have as a motivation for the study.

**Weaknesses:**

The theoretical analysis is based on a strong assumption that the low-dimensional data has a global parameterization. This can limit the application of the results.

The analysis in the paper appears to be standard, and the majority of the idea is adapted from existing research on GANs. However, I would like to emphasize that putting all the details together is not a straightforward task. Moreover, the analysis to MMD is relatively not very common in existing literature. Therefore, I am not criticizing the contribution of the paper, but just pointing out that the technical novelty might not be significant.

I would suggest writing a corollary in Remark 11.

In Section 4, it might be helpful to provide some examples, such as a smooth $d$-dimensional manifold has an upper Minkwoski dimension bounded by $d$. More discussions can be found in "Sharp asymptotic and finite-sample rates of convergence of empirical measures in Wasserstein distance".

**Questions:**

See weakness above.

---

> ### Author Response · Authors · 2023-11-21
> **Response to Reviewer JGBa**
>
> We thank the reviewer for the encouraging remarks and insightful comments. We are glad to note your positive evaluation of the paper's quality of writing, and we appreciate your favorable comments regarding its robustness, clarity, and meaningful contributions.
>
> We agree with the reviewer that the proof techniques presented in this paper are largely built upon the works done in the recent literature. However, as already noted by the reviewer, applying the proof technique, particularly in the MMD context requires additional intricacies and has to be carefully dealt with. Indeed as already pointed out on page 6, A1 is a strong assumption. Certainly, in the context of GANs, a less stringent assumption can be employed since the learning task becomes considerably simpler. As observed by the reviewer, a parallel challenge arose in the analysis of autoencoders, as elucidated by Liu et al. (2023). They addressed this issue by exploring chart-autoencoders, which incorporate additional components in the network architecture compared to conventional autoencoders. We thank the reviewer for the suggestions. Indeed compact $\tilde{d}$-dimensional differentiable manifolds have a Minkowski dimension of at most $\tilde{d}$. Thus, the main result, i.e Theorem 8 and the subsequent hold with $d_\mu$ replaced with $\tilde{d}$, when the data-support is a $\tilde{d}$-dimensional differentiable manifold. A similar result holds for other examples such as affine convex sets, self-similar sets, etc. as highlighted in Proposition 9 of Weed and Bach (2019). We thank the reviewer for stimulating discussion and will include the following discussion in the revised manuscript.
>
> We recall that we call a set $\mathcal{M}$ is $\tilde{d}$-regular w.r.t. the $\tilde{d}$-dimensional Hausdorff measure $\mathbb{H}^{\tilde{d}}$ if $$\mathbb{H}(B_\varrho(x, r)) \asymp r^{\tilde{d}},$$
> for all $x \in \mathcal{M}$ (see Definition 6 of Weed and Bach (2019)). It is known (Mattila, 1999) that if $\mathcal{M}$ is $\tilde{d}$-regular, then the Minkowski dimension of $\mathcal{M}$ is $\tilde{d}$. Thus, when $\text{supp}(\mu)$ is $\tilde{d}$-regular, $d_\mu = \tilde{d}$. Since compact $\tilde{d}$-dimensional differentiable manifolds are $\tilde{d}$-regular (Proposition 9 of Weed and Bach (2019)), this implies that for when $\operatorname{supp}(\mu)$ is a compact differentiable $\tilde{d}$-dimensional manifold, the error rates for the sample estimates scale as in Theorem 8, with $d_\mu$ replaced with  $\tilde{d}$. A similar result holds when $\text{supp}(\mu)$ is a nonempty, compact convex set spanned by an affine space of dimension $\tilde{d}$; the relative boundary of a nonempty, compact convex set of dimension $\tilde{d} + 1$; or a self-similar set with similarity dimension $\tilde{d}$.

---

> > ### Comment · Reviewer_JGBa · 2023-11-23
> >
> > Thank you for the detailed additional discussion. I am happy to keep my score and recommend acceptance.

---

### Official Review · Reviewer_GHGf · 2023-11-01

**Soundness:** 3 good
**Presentation:** 3 good
**Contribution:** 3 good
**Rating:** 6
**Confidence:** 3

**Summary:**

The paper provides a statistical analysis for Wasserstein Autoencoder (WAE) method. Specifically, the paper gives an upper bound for the excess risk for the empirical estimator of the encoder and decoder when both are approximated by neural networks. Under certain assumptions, the proposed method obtains an approximate of the optimal en/decoders, which is shown to have a converging excess risk. The analysis is applied to both Wasserstein and MMD dissimilarities. Furthermore, the adequate latent dimension is used in the bounds, showing that lower dimensional data complies with the corresponding rate.

**Strengths:**

The paper is overall well written and presented, and the the ideas are original to the knowledge of the reviewer. The discussion of all results seem plenty and extensive. Some strengths:
1. The idea of analyzing WAE statistically seems to be an interesting question, as most of such study seem to be on other generative models. The paper provides solid convergence guarantees, which seems expected but still crucial.
2. The paper provides thorough discussion on the error and convergence of the model, including both sample complexity and asymptotics. The error bound accounts for sample size, neural network size, optimization oracle, and dimensionality dependence, which is rather complete, as these includes almost all aspects of concern in practice. The limitations are also adequately addressed.

**Weaknesses:**

1. Though the paper provides the analysis all the way to containing both sample complexity and misspecification, the single question of sample complexity seems to be an interesting question on its own, i.e. population v.s. empirical over the same Holder class, not neural networks. In this setting, optimization is out of the picture, thus questions such as the convergence rate of $\|id - G\circ E\|$ or $\|\hat{G}-G\|$ can be asked without ambiguity. The reviewer understands that this is additional work, but encourages the authors to provide a discussion, rather than stuffing all ingredients all at once. This could possibly be related to the tightness (minimaxity) of the proposed bound.

2. In Remark 11, the construction of $E^*$ does not seem correct, as repeating $E_1$ several times will only result in a graph supported on the diagonal, thus $E^{*}_\sharp\mu$ is not the uniform distribution over the higher dimensional cube. Similarly, the next claim in the remark over the use of $\nu$ with independent marginals seems unjustified.

**Questions:**

Please see above (section Weaknesses) for details.

---

> ### Author Response · Authors · 2023-11-21
> **Response to Reviewer GHGf**
>
> We thank the reviewer for the overall positive feedback of our work. We appreciate the time and effort you invested in reviewing our work. We are pleased to see that you find the paper well written, and we are grateful for the positive remarks on its soundness, presentation, and contribution.
>
> We acknowledge your insightful comments regarding the strengths and weaknesses of our paper. Your positive evaluation of the originality of our ideas and the importance of statistically analyzing Wasserstein Autoencoder (WAE) is encouraging. We also appreciate your recognition of the thorough discussion on error and convergence, as well as the comprehensive consideration of various factors such as sample size, neural network size, optimization oracle, and dimensionality dependence.
>
> Regarding the weaknesses you pointed out, we agree that the single question of sample complexity is interesting on its own. The convergence of $\hat{G}$ to $G$ is guaranteed in Theorem 15, under additional assumptions of uniform equicontinuity. For uniformly Lipshitz generators, the convergence rate is upper bounded by the rates derived in Theorem 8, as stated in Remark 16. Similarly, Proposition 13 deals with $id - G \circ E$. We agree with the reviewer that a more focused discussion on decoding (Proposition 13) and data generation guarantee (Theorem 15) will benefit the readability of the paper. In the final version of the manuscript, we will formally state Remark 16 as a corollary for further clarification and streamline Section 5.5 to predominantly cover these two aspects. In response to your thoughtful feedback, we will omit Remark 11 as it diverges from the primary theme of the paper.

---

> > ### Comment · Reviewer_GHGf · 2023-11-21
> > **Thanks for the response**
> >
> > I thank the authors for the response, which addresses all my questions. I thus leave my rating unchanged.

---

### Official Review · Reviewer_oqXa · 2023-11-06

**Soundness:** 3 good
**Presentation:** 3 good
**Contribution:** 3 good
**Rating:** 6
**Confidence:** 4

**Summary:**

The paper introduces a comprehensive framework for analyzing error rates in learning unknown distributions through Wasserstein Autoencoders (WAEs), particularly in scenarios where data points exhibit an intrinsic low-dimensional structure within a high-dimensional feature space. By characterizing this dimensionality using the Minkowski dimension of the target distribution's support, the paper establishes an oracle inequality that effectively balances model-misspecification and stochastic errors, offering insights into the trade-offs achievable through appropriate network architectures. Furthermore, the framework allows for a thorough examination of the accuracy of encoding and decoding guarantees, as well as the approximation of the target distribution by the generated push-forward measure.
While the paper offers valuable insights into the theoretical underpinnings of WAEs, it acknowledges the challenges associated with accurately estimating the complete error due to the complexities of the optimization process. Notably, the paper emphasizes the need for further analysis, especially in contexts involving non-deterministic network outputs and Gaussian-based latent distributions, which are commonly employed in practical applications.

**Strengths:**

The paper demonstrates several strengths, particularly in its provision of theoretical analyses and implications regarding the selection of network sizes for encoders and decoders based on the number of training data samples. Notably, the theoretical results, particularly Theorem 8, provide valuable insights that ensure encoding and decoding guarantees, highlighting the paper's ability to establish a close relationship between the encoded distribution and the target latent distribution. Furthermore, the paper emphasizes the effectiveness of the generator in accurately mapping the encoded points back to their original positions, showcasing its robust understanding of the intricacies involved in data transformation and representation within the context of the study.

**Weaknesses:**

The paper exhibits certain weaknesses that should be addressed to strengthen its overall contribution. Firstly, while the paper presents intriguing theoretical results, the absence of an experimental demonstration, even with synthetic datasets, limits its practical applicability and hinders the validation of the theoretical findings in real-world scenarios. The lack of empirical evidence to support the theoretical claims diminishes the paper's credibility and inhibits a comprehensive understanding of the practical implications of the proposed concepts. Additionally, the paper falls short in elucidating how the theoretical results can be practically employed to guide the selection or design of encoder/decoder networks, particularly for enhancing the generation of sharper and more realistic images within the context of Wasserstein Autoencoders (WAEs). The absence of clear guidelines or methodologies for leveraging the theoretical findings to improve the image generation process highlights a critical gap in the paper's applicability and its potential impact on practical applications within the field. Addressing these limitations would significantly enhance the paper's overall credibility and ensure its relevance and practical significance within the research domain.

**Questions:**

No question

---

> ### Author Response · Authors · 2023-11-21
> **Response to Reviewer oqXa**
>
> We thank the reviewer for their insightful comments and are glad to hear feedback that the work is significant, with an insightful theoretical contribution extending previous work, and has a potential impact on understanding the intricacies of WAEs.  The theoretical results, particularly Theorem 8, are indeed crucial in providing a deep understanding of encoding and decoding guarantees, emphasizing the connection between the encoded distribution and the target latent distribution. We are pleased that you find value in our insights into encoding and decoding guarantees, as well as the effectiveness of the generator in accurately mapping encoded points back to their original positions.
>
>  We value your constructive critique of our paper. The empirical evidence presented in Section 2 illustrates a polynomial-like decay pattern as hinted at by Theorem 8 and its corollaries. It is important to highlight that such empirical validation is notably absent in the existing literature on GANs or WAEs including the very recent works of  Chakrabarty and Das (2021), Dahal et al. (2022), and Huang et al. (2022). The primary contribution of our paper lies in comprehending how the convergence rates of sample estimates rely solely on the intrinsic dimension of the data. In this regard, the experiment in Section 2 demonstrates that the error rate of sample estimates decays at a rate depending on the Minkowski dimension of the data. In this regard, Theorem 8 provides a recommendation for the appropriate network size based on the number of samples and the underlying distribution. However, practically, the population quantities, as assumed in A1, remain unknown, making it challenging to ascertain whether the derived rates are optimal in a minimax sense. In response to the reviewer's recommendation, we will broaden the experimental study to encompass additional datasets to show that a similar trend as shown in Fig. 1, holds across multiple datasets. This expansion will enrich the empirical validation and strengthen the practical implications of our findings. To maintain conciseness, we intend to include this expanded analysis in the supplement for space efficiency.

---

### Meta-Review · Area_Chair_W3iY · 2023-12-05

**Metareview:**

This is a borderline paper and the reviewers highlight various minor concerns that should be addressed in a final version. Overall, however, they propose to accept the paper and I'm following their recommendation.

**Justification For Why Not Higher Score:**

It is a borderline paper which lies between accept and reject. It is definitely not a spotlight paper.

**Justification For Why Not Lower Score:**

A lower score is possible but the reviewers are quite clear in their recommendation that they would like to see the paper accepted.

---

### Decision · Program_Chairs · 2024-01-16

Accept (poster)